# *Plasmodium* blood stage development requires the chromatin remodeller Snf2L

Maria Theresia Watzlowik[1], Elisabeth Silberhorn[1], Sujaan Das[2], Ritwik Singhal[3], Kannan Venugopal[4,5], Simon Holzinger[1], Barbara Stokes[4,5], Ella Schadt[2], Lauriane Sollelis[4,5], Victoria A. Bonnell[3], Matthew Gow[2], Andreas Klingl[6], Matthias Marti[4,5], Manuel Llinás[3,7], Markus Meissner[2✉] & Gernot Längst[1✉]

The complex life cycle of the malaria parasite *Plasmodium falciparum* involves several major differentiation stages, each requiring strict control of gene expression. Fundamental changes in chromatin structure and epigenetic modifications during life cycle progression suggest a central role for these mechanisms in regulating the transcriptional program of malaria parasite development[1–6]. *P. falciparum* chromatin is distinct from other eukaryotes, with an extraordinarily high AT content (>80%)[7] and highly divergent histones resulting in atypical DNA packaging properties[8]. Moreover, the chromatin remodellers that are critical for shaping chromatin structure are not conserved and are unexplored in *P. falciparum*. Here we identify *P. falciparum* Snf2L (*Pf*Snf2L, encoded by *PF3D7_1104200*) as an ISWI-related ATPase that actively repositions *P. falciparum* nucleosomes in vitro. Our results demonstrate that *Pf*Snf2L is essential, regulating both asexual development and sexual differentiation. *Pf*Snf2L globally controls just-in-time transcription by spatiotemporally determining nucleosome positioning at the promoters of stage-specific genes. The unique sequence and functional properties of *Pf*Snf2L led to the identification of an inhibitor that specifically kills *P. falciparum* and phenocopies the loss of correct gene expression timing. The inhibitor represents a new class of antimalarial transmission-blocking drugs, inhibiting gametocyte formation.

The *P. falciparum* life cycle and its various developmental stages require a tightly orchestrated transcriptional program. However, the full range of variation in gene expression cannot solely be explained by stage-specific transcription factors as they are highly under-represented in the *P. falciparum* genome[9–13]. The chromatin structure at promoters either restricts or enables transcription factor access, thereby representing a crucial mechanism for gene expression regulation in eukaryotes. Histone octamers positioned on DNA are a major obstacle for sequence-specific DNA-binding factors, and eukaryotic epigenetic regulatory mechanisms such as site-specific DNA/histone modification or RNA-mediated processes also impact chromatin accessibility[14–16]. Furthermore, the maintenance and re-establishment of chromatin after transcription elongation or DNA replication require molecular machinery for nucleosome assembly, positioning and spacing[15,17,18]. Chromatin-remodelling enzymes (CREs) are ATP dependent, exerting movement, disruption or assembly of nucleosomes, the exchange of histones for variants, or generally altering chromatin structure to control access to DNA in chromatin[19–21]. Histone CREs and their associated proteins are highly conserved across eukaryotes. Human cells have evolved 53 Snf2 family enzymes, forming more than 1,000 distinct multiprotein complexes with their associated proteins[22,23]. By contrast, *P. falciparum* encodes only ten enzymes with related ATPase domains but they lack the crucial nucleosome remodelling motifs[13,24]. Moreover, known CRE-associated complex subunits are not present in *P. falciparum*. It is therefore unclear whether these *P. falciparum* ATPases represent functional remodelling enzymes and how they integrate into multiprotein complexes. The enzyme divergence may represent an adaptation to the extraordinarily high AT content, which averages over 90% in non-coding regions, and to the highly divergent chromatin architecture, suggesting different mechanisms of *P. falciparum* chromatin organization.

## *Pf*Snf2L is an active CRE

Here we explored the functional role of *Pf*Snf2L, which is most related to the ATP-dependent ISWI subfamily of CREs[25], but exhibits only 30% homology to the human SNF2L (*Hs*SNF2L; encoded by *SMARCA1*) (Extended Data Fig. 1a,b). The protein contains a relatively conserved ATPase core but lacks conservation in the autoregulatory and substrate-binding domains (Fig. 1a). The missing domains and large asparagine-rich insertions (Extended Data Fig. 1c,d) bring into question its suggested activity as a nucleosome-remodelling machine. Using recombinantly purified *Pf*Snf2L, we find that this

[1]Regensburg Center for Biochemistry (RCB), University of Regensburg, Regensburg, Germany. [2]Experimental Parasitology, Department of Veterinary Sciences, Faculty of Veterinary Medicine, Ludwig-Maximilians-University, Munich, Germany. [3]Department of Biochemistry and Molecular Biology and Huck Center for Malaria Research, Pennsylvania State University, State College, PA, USA. [4]Institute of Parasitology, Vetsuisse and Medical faculty, University of Zurich, Zurich, Switzerland. [5]Institute of Infection and Immunity, College of Medical, Veterinary and Life Sciences, University of Glasgow, Glasgow, UK. [6]Plant Development, Ludwig-Maximilians-University Munich, Planegg-Martinsried, Germany. [7]Department of Chemistry, Pennsylvania State University, State College, PA, USA. ✉e-mail: markus.meissner@lmu.de; gernot.laengst@ur.de

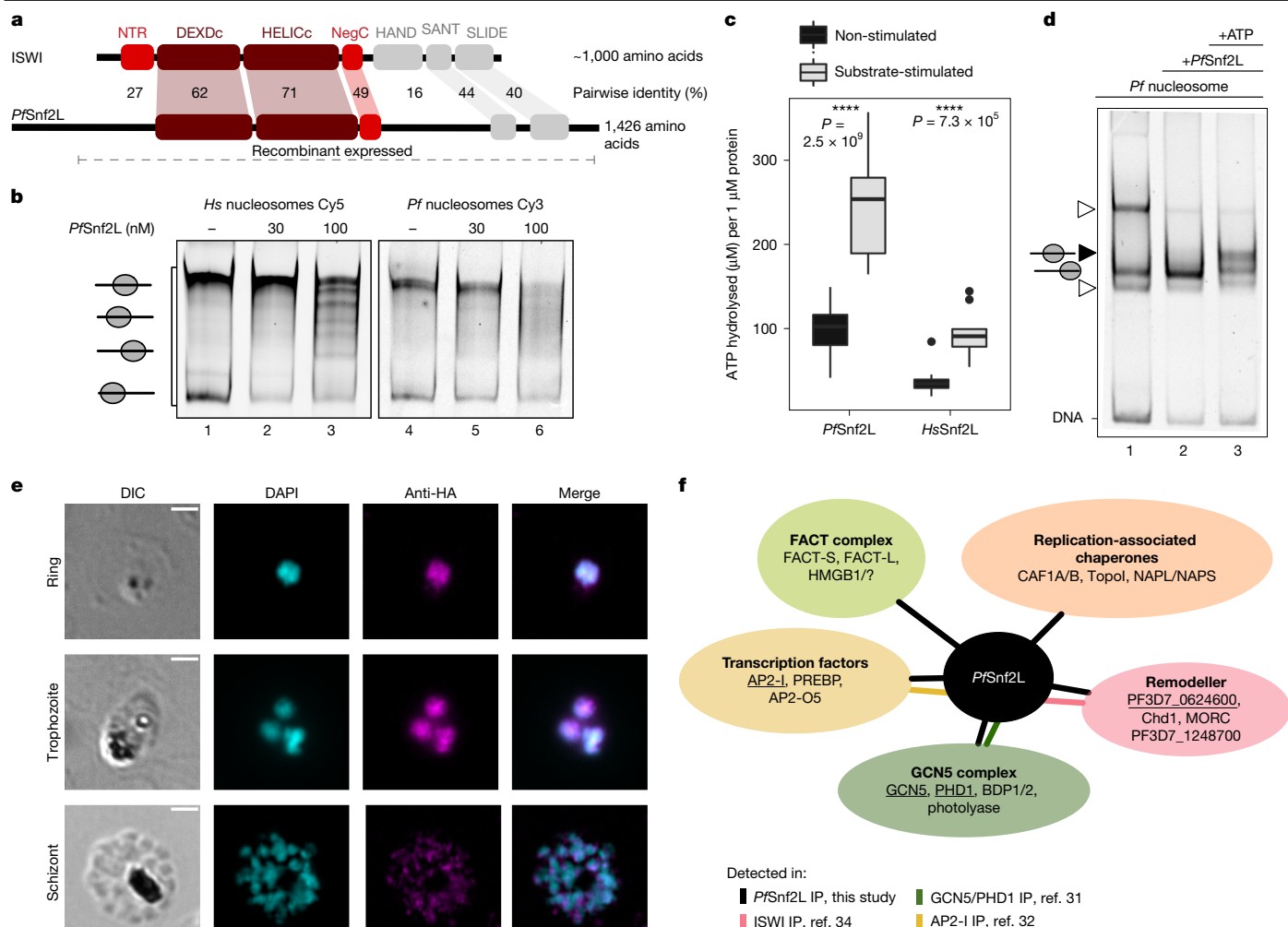

**Fig. 1 | Enzymatic activity and interaction partners of the divergent chromatin remodeller _Pf_Snf2L. a**, Comparison of typical ISWI-domain architecture with _Pf_Snf2L. The location of ATPase regions (dark red), autoregulatory domains (light red) and HAND–SANT–SLIDE domains (grey) and their degree of conservation are indicated. **b**, Competitive nucleosome remodelling assay using recombinant _Pf_Snf2L, ATP and nucleosomal templates with Cy3- and Cy5-labelled DNA, either reconstituted with recombinant _P. falciparum_ or recombinant human histone octamers, as indicated. The DNA template has a central nucleosome-positioning sequence (NPS), flanked by 77 bp of linker DNA on both sides. **c**, The non-stimulated or nucleosome-stimulated ATP hydrolysis rate of _Pf_Snf2L/_Hs_SNF2L. $n = 18$ and $n = 10$ independent experiments. The centre central line represents the median, the box limits

span the interquartile range (IQR), the whiskers extend to 1.5× the IQR and the dots represent outliers beyond this range. Statistical analysis was performed using unpaired two-tailed Student's _t_-tests; ****$P < 0.001$. **d**, Nucleosome assembly assay using _Pf_Snf2L and canonical _P. falciparum_ histones. Nucleosomes and uncharacterized histone–DNA complexes (triangle) are indicated. **e**, _Pf_Snf2L–HA colocalization IFAs in asexual blood stages. DIC, differential interference contrast; DAPI, 4′,6-diamidino-2-phenylindole. Scale bars, 2 μm. One representative image is shown of $n = 3$ independent experiments. **f**, Interaction network showing proteins that were identified by _Pf_Snf2L IP–LC–MS/MS (black lines). Confirmed interactions detected in complementary IP experiments (bait protein is underlined) are indicated by connectors in the corresponding colours.

divergent enzyme preferentially binds to AT-rich DNA and specifically to nucleosomes (Extended Data Fig. 2a–g). _Pf_Snf2L also moves histone octamers along DNA in an ATP-dependent manner, and this activity is equally strong on either recombinant human or _Plasmodium_ nucleosome substrates (Fig. 1b and Extended Data Fig. 2h–l). _Pf_Snf2L is a nucleosome-stimulated ATPase-like ISWI enzyme with an increased ATP hydrolysis rate and reduced remodelling efficacy relative to _hs_SNF2L (Fig. 1c and Extended Data Fig. 2h,i), potentially due to the missing substrate-recognition domains[26–28]. Furthermore, _Pf_Snf2L can reassemble octamers from a DNA–histone mixture into well-positioned nucleosomes in the absence of ATP (Fig. 1d).

Endogenous C-terminal tagging of _Pf_Snf2L in 3D7 parasites reveals its expression during all asexual blood stages, peaking in the late-ring/trophozoite stage (Extended Data Fig. 3a,b). Immunofluorescence analysis (IFA) of _Pf_Snf2L showed nuclear localization with uneven distributed in the nucleoplasm and spots of local enrichment (Fig. 1e).

_Pf_Snf2L immunoprecipitation (IP) combined with liquid chromatography–tandem mass spectrometry (LC–MS/MS) analysis identifies specific interactions with proteins associated with chromatin organization (Extended Data Fig. 3c,d and Supplementary Table 1). In total, we detected 21 Apicomplexan- or _Plasmodium_-specific proteins, including several uncharacterized proteins. Neither the IP experiment nor in silico attempts identify orthologues of characteristic subunits of ISWI-type complexes in _P. falciparum_, pointing towards the existence of novel, divergent CRE complexes. Some detected interactors are associated with transcriptional regulation and chromatin assembly, remodelling and organization (Fig. 1f). For example, the FACT complex has an important role in nucleosome assembly during transcription elongation[29], while CAF and NAP proteins function in replication-coupled assembly[30]. Other likely interactors are one or more versions of the GCN5 complex, as multiple GCN5 complex subunits are detected in the _Pf_Snf2L IP, and their interaction has been confirmed in a GCN5 IP

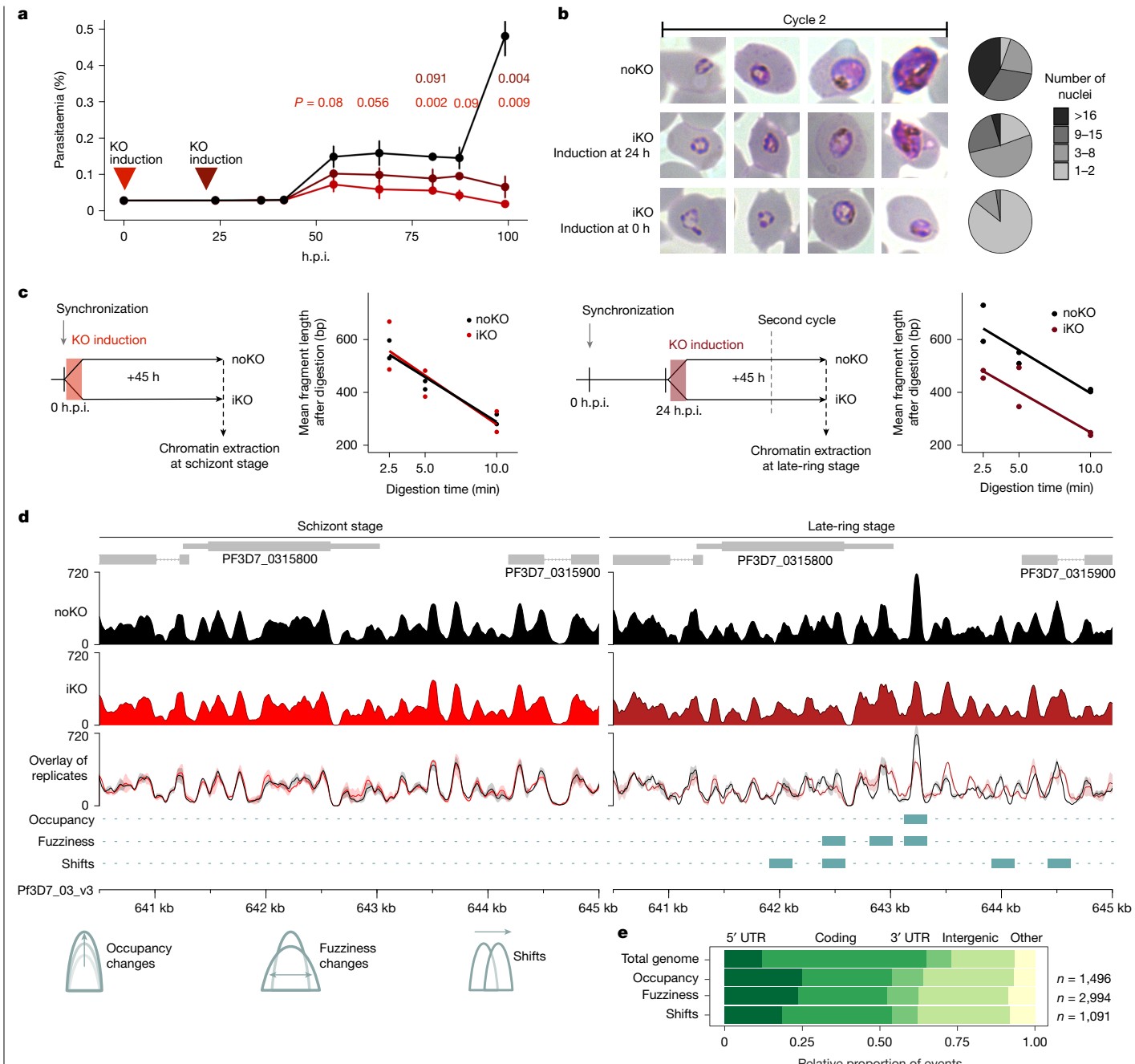

**Fig. 2 | Pf*Snf2L* KO results in chromatin structure changes at the ring stage, preventing parasite development and viability. a**, Growth-curve analysis of synchronized *Pf*Snf2L–HA parasites that were not induced (black), induced at 24 h.p.i. (dark red) and induced at 0 h.p.i. (light red). Data are mean ± s.d. of three technical replicates (*n* > 100 per timepoint). noKO is the reference group. Statistical analysis was performed using unpaired two-tailed Student's *t*-tests; not significant (NS): *P* > 0.05, **P* < 0.05, ***P* < 0.01, ****P* < 0.005. **b**, Representative Giemsa-stained blood smears of parasites treated as in **a**. The pie charts show the number of nuclei at the end of cycle 2. *n* = 100 each. **c**, Schematic of KO induction and chromatin extraction at the schizont (left) and late-ring (right) stages. The MNase digestion kinetics are presented accordingly. **d**, Exemplary coverage plots of MNase-seq data of schizont parasites (mean of triplicates) and late-ring parasites (mean of duplicates) after Pf*Snf2L* KO. The range between the minimum and maximum values of replicates is indicated by the shaded area in the overlay. Identified changes in occupancy, fuzziness and shifts (Supplementary Note 1) are indicated. The region shown is chromosome 3:641000–645000 with annotated genes marked in grey. **e**, Assignment of Pf*Snf2L*-KO-dependent nucleosome positioning changes relative to functional elements of the genome. *n* indicates the number of nucleosomal changes.

study[31]. These results reveal a role for *Pf*Snf2L in nucleosome assembly and imply chromatin-dependent regulation of gene expression[31–34].

## Essential role of *Pf*Snf2L for asexual development

To examine the biological function of *Pf*Snf2L in parasites, we used an inducible dimerizable *cre*–*lox* (DiCre)-mediated gene excision system[35].

Efficient induction of gene excision (>95% induction rate) and resulting depletion of *Pf*Snf2L was verified at the DNA, mRNA and finally protein level around 46 h after addition of rapamycin (Extended Data Fig. 3e–i). To analyse the phenotypic consequences of *Pf*Snf2L depletion, we induced the knockout (KO) in synchronized parasites at 0 h and 24 h post-invasion (h.p.i.). In both cases, development and replication are impaired in the second cycle before parasite death (Fig. 2a,b).

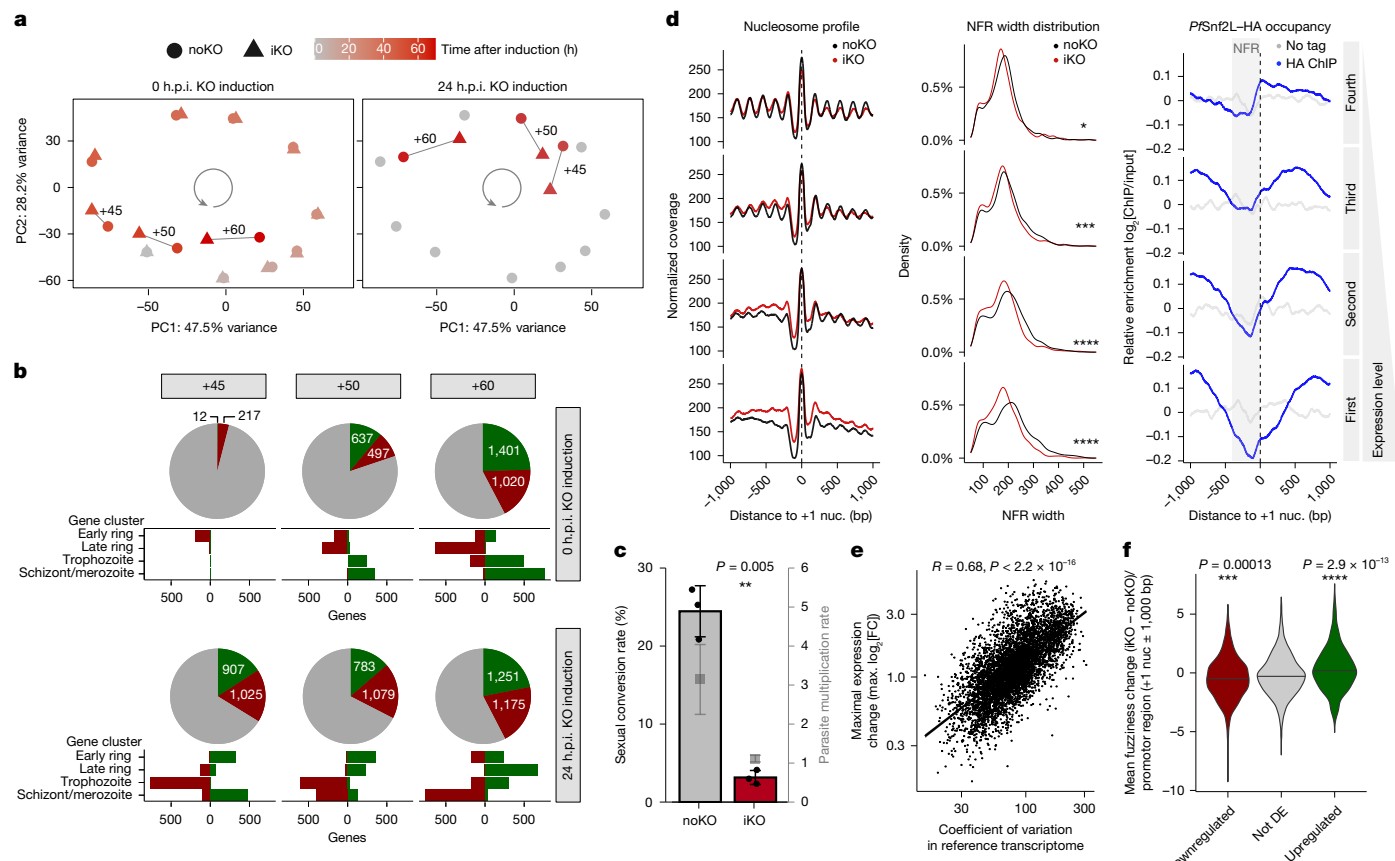

**Fig. 3 | *Pf*Snf2L regulates just-in-time transcription by shaping the promoter architecture of stage-specific genes. a**, Principal component (PC) analysis of whole RNA sequencing (RNA-seq) data of highly synchronous noKO (circle)/ iKO (triangle) parasites at different timepoints (45 h, 50 h and 60 h) after KO induction at 0 h (left) or 24 h (right). Data are the mean of triplicates. **b**, The numbers of DEGs (activated (green), repressed (red)) in total and among the four stage-specific gene clusters (as in Extended Data Fig. 8b) over time in iKO and noKO parasites as in **a**. **c**, Sexual conversion rates after *Pf*Snf2L KO induced at 10 h.p.i. before gametocyte induction at 24 h.p.i. Single datapoints and the mean ± s.d. of triplicates are shown. Statistical analysis was performed using paired two-tailed Student's *t*-tests. PMRs on the right axis are shown in grey

(mean ± s.d.). **d**, MNase-seq occupancy profiles of late-ring-stage iKO/noKO parasites around the +1 nucleosome (nuc.) (left; mean of duplicates), NFR width distribution (middle; mean of duplicate) and *Pf*Snf2L enrichment (chromatin immunoprecipitation followed by sequencing (ChIP–seq) at 34 h.p.i.; right; mean of duplicates), subdivided by gene expression quantiles **e**, Correlation between the maximal expression change of each gene and its expression variation in the reference transcriptome as in **a**. Statistical analysis was performed using the two-sided Pearson test. FC, fold change. **f**, The mean nucleosome fuzziness difference (iKO − noKO) of all promoter regions (±1,000 bp around the +1 nucleosome) with respect to differential expression. DE, differentially expressed. Statistical analysis was performed using unpaired two-tailed Student's *t*-tests.

Notably, early induction (at 0 h) results in defects in merozoite release during host cell exit, as shown by live imaging (Supplementary Videos 1 and 2), resulting in a reduced rate of newly infected red blood cells (RBCs) (Extended Data Fig. 4a–d). Until egress, the number of nuclei, schizont segmentation, cytogenesis and invasion organelle formation are unaffected (Extended Data Fig. 4e–g). Staining of the host cell during egress indicates modified RBC properties at this specific timepoint in KO-induced (iKO) parasites compared with in non-induced (noKO) parasites (Extended Data Fig. 4h). The developmental defect in the second cycle is similarly observable when KO is induced at 24 h.p.i., but shifted in time, correspondingly (Fig. 2a,b). Taken together, these data reveal that *Pf*Snf2L has an essential role in parasite survival and in driving the development of asexual stages.

To determine whether the substantial developmental effects are caused by perturbation of the parasite chromatin landscape, we used micrococcal nuclease (MNase) digestion and sequencing (MNase-seq) of schizont or late-ring chromatin with or without Pf*Snf2L* KO (Fig. 2c). The MNase-seq method enables both the detection of individual nucleosome positions and monitoring of changes in the general accessibility of chromatin[36]. MNase digestion kinetics at the schizont stage showed no detectable effect on chromatin accessibility, with noKO and iKO parasites having a similar time-dependent DNA length

pattern (Extended Data Fig. 5a). However, after invasion, at the late-ring stage, the chromatin of iKO parasites is more nuclease-accessible than in the noKO parasites, releasing mononucleosomal DNA with lower MNase activity (Fig. 2c). Thus, the loss of *Pf*Snf2L induces a global opening of chromatin at the ring stage. To investigate the local chromatin structure and individual nucleosome positions, noKO and iKO chromatin—digested to a similar degree—were analysed in the schizont and ring stages. Only minor changes in the nucleosome maps can be observed in schizonts (<1%). However, we found substantial changes in nucleosome positioning at late ring stages (exemplary tracks are shown in Fig. 2d and Extended Data Fig. 5b), where approximately 20% of the well-positioned nucleosomes are transformed (Extended Data Fig. 5c,d and Supplementary Note 1). Changes in nucleosome occupancy, fuzziness and positioning are enriched in the 5′ untranslated region (UTR) and intergenic regions (Fig. 2e). These regions are associated with regulatory processes, comprising promoters with a highly defined chromatin architecture.

## *Pf*Snf2L is a just-in-time regulator of gene expression

Substantial changes in the chromatin structure at regulatory elements after Pf*Snf2L* KO could result in downstream effects on gene

expression that may lead to the observed developmental defects. Time-resolved transcriptomic analysis of KO parasites reveals a delayed regulation of gene expression, correlating with the disappearance of *Pf*Snf2L (Extended Data Fig. 6a,b). Most *P. falciparum* genes show a stage-dependent expression pattern, being activated or repressed at specific timepoints of the erythrocytic life cycle. These genes were grouped into four stage-specific gene clusters, showing expression peaks at the early-ring (1,417 genes), late-ring (1,226 genes), trophozoite (1,421 genes) or schizont/merozoite (1,389 genes) stages (Extended Data Fig. 6c and Supplementary Table 3). The loss of *Pf*Snf2L leads to delayed activation of genes being activated in this respective stage, and genes being turned off in this stage are similarly delayed in their repression (Extended Data Fig. 6d). These changes in timing fidelity globally slow down the cyclic trajectory of gene expression during intraerythrocytic development, with the first effects being visible approximately 40 h after KO induction (Fig. 3a and Extended Data Fig. 6b). Considering all stages throughout the asexual blood cycle, distinct stage-specific sets of genes are differentially expressed (DEGs) depending on the early (0 h) or late (24 h) induction of KO, with about 200 DEGs in the schizont stage, mainly early-ring-specific genes, and around 2,000 in the ring and trophozoite stages (Fig. 3b). The data show that Pf*Snf2L* KO does not affect the expression of specific gene categories, but a loss of the remodelling enzyme rather impacts genes undergoing activation or repression at a given developmental stage. This indicates that *Pf*Snf2L is required to maintain proper timing of regulation of the entire parasite transcriptome. The gene expression changes are consistent with the phenotype of delayed cell cycle progression. Moreover, the observed egress defect could be explained by the numerous exported proteins downregulated at the schizont stage. The collective knockdown of the multigene exported protein family (EPF) was previously shown to result in inefficient merozoite release, similar to the iKO of *Pf*Snf2L[37].

Notably, in the KO group, genes normally expressed after the asexual blood stages are upregulated in early rings, where they are generally silenced (Supplementary Table 2). Among these misactivated genes are numerous gametocyte-specific genes, which are mostly downregulated in the trophozoite stage (Extended Data Fig. 6e). Genes dysregulated by the lack of *Pf*Snf2L include known epigenetic and transcriptional regulators of sexual development (Extended Data Fig. 6f). Depletion of *Pf*Snf2L before experimental induction of sexual commitment (that is, the erythrocytic parasite state that deterministically results in conversion to the sexual stage) results in a substantially reduced proportion of parasites producing transmission-competent sexual stages (Fig. 3c and Extended Data Fig. 9b), demonstrating that *Pf*Snf2L is crucial for gametocytogenesis. This finding is supported by previous transcriptomic studies of gametocytogenesis, which identified elevated expression levels of *Pf*Snf2L in sexually committed parasites[38,39]. These results demonstrate that *Pf*Snf2L is an important organizer of the nucleosomal landscape at gene promoters and that its activity is required for just-in-time transcription activation/repression throughout the asexual and sexual intraerythrocytic stages of *P. falciparum* development.

## Maintenance of global chromatin structure by *Pf*Snf2L

Regulation of gene expression occurs at multiple levels including transcriptional control, splicing and RNA degradation, with the promoter being a major determinant of transcription. Promoter chromatin architecture, with positioned nucleosomes around a nucleosome-free region (NFR), controls accessibility for transcription initiation factors and transcriptional activity[15]. As we observe the largest KO-dependent changes in chromatin structure in 5′ UTRs and intergenic regions, we examined nucleosome position changes at gene regulatory elements (Extended Data Fig. 5e). Owing to different published annotations of transcription start sites[40–42], we instead used an alternative chromatin-based

annotation method. The NucDyn pipeline[43] was applied to identify the characteristic +1 nucleosome of promoters next to the proposed transcription start site. The alignment of MNase-seq data relative to the well-positioned +1 nucleosome revealed the biggest changes in chromatin structure immediately upstream of +1 (Extended Data Fig. 5f). Subdivision of the data into gene expression quantiles shows the most pronounced effect of Pf*Snf2L* KO on highly expressed genes (first quantile), resulting in a loss of nucleosome positioning (Fig. 3d (left)). Furthermore, NFRs tend to be extended and more accessible with increased expression in the noKO samples, but not in the iKO group, suggesting that *Pf*Snf2L actively opens the NFRs (Fig. 3d (middle)). Consistent with these findings, the *Pf*Snf2L-binding pattern at promoters changes with increasing gene expression levels, showing higher *Pf*Snf2L occupancy surrounding the NFR (Fig. 3d (right)).

Highly expressed genes show the biggest KO-dependent changes in promoter architecture, and these genes also have the biggest KO-dependent changes in mRNA levels. In particular, genes that undergo strong activation or repression during the asexual cycle are *Pf*Snf2L responsive, like the clonally variant genes that are fundamental for adaptation within the host cell environment (Fig. 3e and Extended Data Fig. 6d). Upregulation in Pf*Snf2L* KO coincides with a loss in nucleosome positioning at promoter regions, shown by increased nucleosome fuzziness (Fig. 3f). The opposite effect is observed for downregulated genes (Fig. 3f). We propose that *Pf*Snf2L coordinates promoter architecture with gene expression timing. ISWI-type CREs in other organisms have also been shown to facilitate transcription initiation by enhancing the accessibility of *trans*-acting factors to gene promoters[44]. Furthermore, we find a direct interaction of *Pf*Snf2L with transcription factors and the Gcn5 complex, linking the processes of epigenetic and transcriptional regulation to *Pf*Snf2L function. In support, deletion of the GCN5/PHD1 domain has been shown to result in a delay in developmental progression, with reduced cycle transition and upregulation of non-blood stage-specific genes[31,45], as seen for Pf*Snf2L* KO.

## The anti-plasmodial drug NH125 inhibits *Pf*Snf2L

The essential role of *Pf*Snf2L in the malaria parasite, combined with its high sequence divergence from the human enzymes, suggests that it is a potential drug target. On the basis of its enzymatic activity, a small-molecule library was screened for inhibition and binding to the recombinant ATPase domain. The screening pipeline combines an ATPase activity screen, followed by biophysical analysis of protein–small molecule interactions and in vitro nucleosome remodelling assays (Fig. 4a). Effective inhibitors were validated in vivo, and the toxicity and specificity of the compounds were assessed. The anti-plasmodial chemical compound with the most pronounced effect was NH125 (Extended Data Fig. 7). Exposure of parasites to this compound reproduces the Pf*Snf2L*-KO phenotype, resulting in a similar developmental delay and parasite death as shown in Giemsa staining (Fig. 4b). Furthermore, time-resolved transcriptomic analysis after treatment with 1 μM NH125 shows a highly significant correlation of DEGs when compared to the KO (Fig. 4c,d, Extended Data Fig. 7f and Supplementary Table 2), confirming the direct and specific targeting of *Pf*Snf2L. Biophysical analysis using microscale thermophoresis and initial-fluorescence assays revealed specific, high-affinity binding of NH125 to *Pf*Snf2L (Extended Data Fig. 8a and Supplementary Note 2). Using nano differential scanning fluorometry and dynamic light scattering to assess protein stability and size, we show that NH125 only weakly interacts with *Plasmodium vivax* Snf2L (>20 μM) and does not interact with the *Hs*SNF2L enzyme (Extended Data Fig. 8b). Besides the high specificity of the compound, we show that NH125 binds outside of the ATPase domain, still allowing ATP and DNA binding, but eliminating the ATPase activity of *Pf*Snf2L. We suggest an NH125-induced conformational change that results in the specific aggregation of *Pf*Snf2L (Extended Data Fig. 8c and Supplementary Note 2).

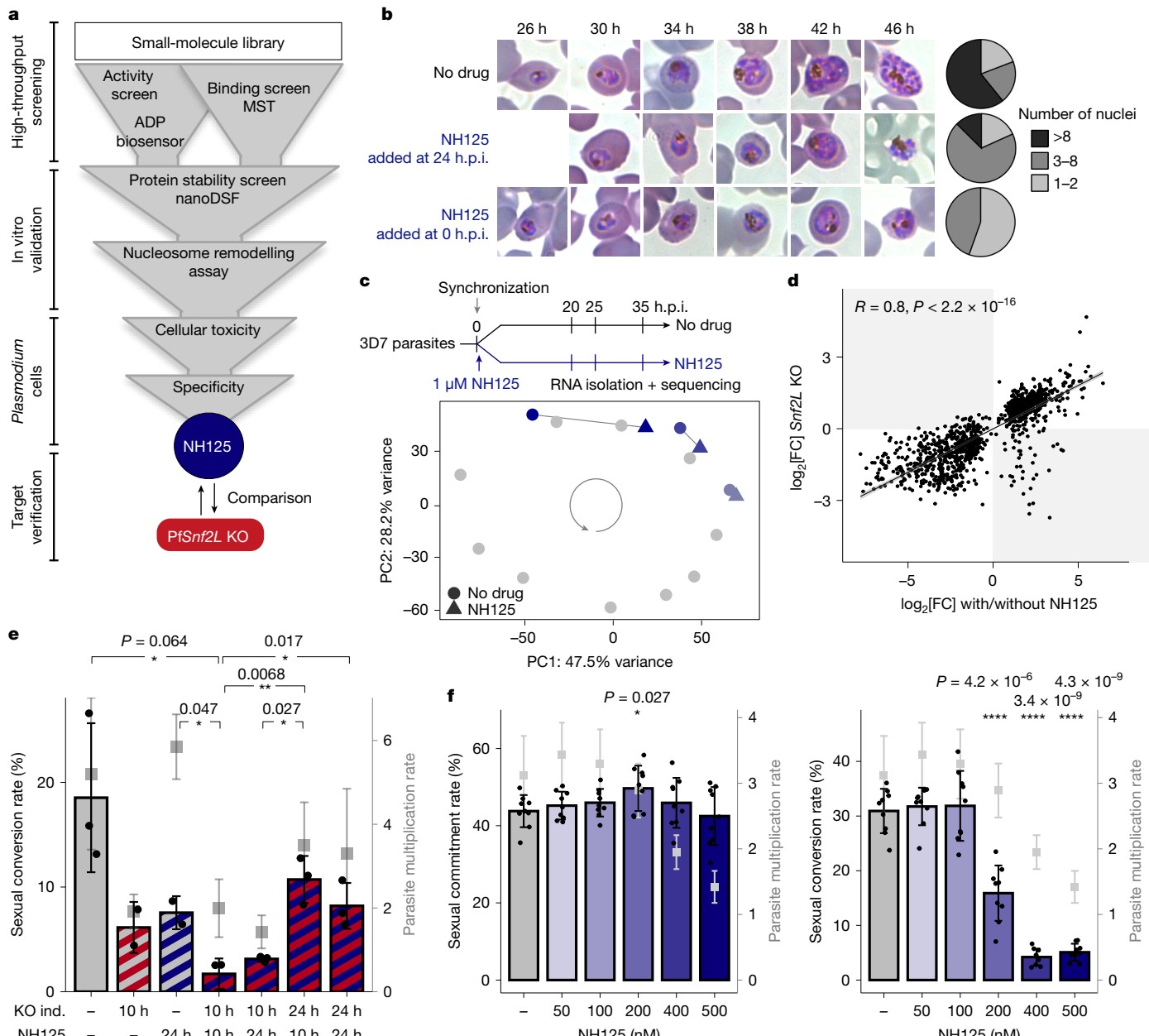

**Fig. 4 | The potent drug NH125 specifically inhibits *Pf*Snf2L and phenocopies KO effects. a**, Schematic of the inhibitor-screening pipeline. MST, microscale thermophoresis; nanoDSF, nano-differential scanning fluorimetry. **b**, Representative Giemsa-stained blood smears of parasites cultured in the presence or absence of 1 μM NH125 added at 0 or 24 h.p.i. The pie charts show the number of nuclei at the end of the cycle. $n > 80$ each. **c**, Schematic of drug treatment and RNA extraction timing (top). Bottom, principal component analysis of whole-transcriptome data (blue) and the reference transcriptome (grey; as in Fig. 3a). Data are the mean of triplicates. **d**, Correlation between expression change in trophozoites after *Pf*Snf2L KO and NH125 treatment for DEGs ($P < 0.01$, two-sided Pearson test). **e**, The sexual conversion rate after *Pf*Snf2L KO induction (ind.) (red) and/or NH125 treatment (blue) at 10 and 24 h.p.i. Single datapoints and mean ± s.d. of triplicates are shown. PMRs on the right axis are shown in grey (mean ± s.d.). Statistical analysis was performed using unpaired two-tailed Student's *t*-tests. **f**, Simultaneous readout of sexual commitment rate (left) and sexual conversion rate (right) of the dual-reporter NF54 parasites after NH125 treatment at 10 h.p.i. Single datapoints and mean ± s.d. of triplicates are shown. PMRs on the right axis are shown in grey (mean ± s.d.). Statistical analysis was performed using unpaired two-tailed Student's *t*-tests.

## *Pf*Snf2L is required for early gametocytogenesis

Notably, in addition to killing the parasites in the asexual erythrocytic cycle, the sexual conversion rate was substantially decreased after just one day of drug treatment in a dose-dependent manner, confirming the proposed function of *Pf*Snf2L in activating the sexual stage of the parasite (Supplementary Note 2 and Extended Data Fig. 9c). The reduced ability to form gametocytes after *Pf*Snf2L-KO was reinforced by combining rapamycin and NH125. This complete depletion

of *Pf*Snf2L leads to substantially impaired sexual conversion that is more pronounced than after single treatment (Fig. 4e and Extended Data Fig. 9d). To differentiate whether sexual commitment or conversion is affected during gametocytogenesis, we used a dual-reporter line (Extended Data Fig. 10 and Supplementary Note 2) that enables independent quantification of the two processes in the same experiment. Consistent with the role of *Pf*Snf2L in regulating just-in-time transcription, sexual commitment is not affected at 200 nM NH125. By contrast, the sexual conversion rate is significantly reduced at this

concentration, indicating impaired regulation of early gametocyte development (Fig. 4f and Extended Data Fig. 9e,f). This effect is not due to general NH125-induced toxicity, as the parasite multiplication rate (PMR) remains unaffected under these experimental conditions (Fig. 4f and Extended Data Fig. 9e,f).

## Discussion

*Pf*Snf2L is an essential epigenetic regulator, driving developmental progress by controlling the timely activation and repression of stage-specific genes, and early regulation of gametocytogenesis. Confirming the importance of CREs, a related Snf2-subfamily member was identified and confirmed to regulate late gametocyte differentiation in *P. berghei*[46]. We propose that the CRE–transcription factor interplay tunes the enzymatic activity and nucleosome positioning behaviour of *Pf*Snf2L. KO parasites display substantial changes in chromatin architecture, affecting the promoters of stage-specific genes and thereby regulating the spatiotemporal loading of *trans*-acting factors onto target genes.

*Pf*Snf2L is an ATP-dependent nucleosome-remodelling enzyme that lacks sequence conservation with other eukaryotic orthologues and interacts with a novel set of chromatin-related interactors. The functional and architectural divergence may have co-evolved with the *P. falciparum*-specific chromatin features, like adaptation to a genome with high AT content and rather unstable nucleosomes[8].

*Pf*Snf2L's essential role and sequence divergence make it a potential drug target. The *P. falciparum*-specific small-molecule inhibitor NH125 inhibits its activity, substantially interfering with intraerythrocytic cycle progression and gametocytogenesis, resulting in parasite death. NH125 presents a solid lead for the further development of a new class of antimalarial transmission-blocking drugs.

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

## Methods

### Phylogenetic analysis and multiple sequence alignments

A list of the sequences of CREs used for phylogenetic analysis is provided in Supplementary Table 3. Phylogenetic trees were generated using Geneious Tree Builder (Geneious software v.11.1.5): global alignment with free end gaps; Blosum62 as cost matrix; Jukes–Cantor distance model; method neighbor-joining; gap open penalty 10; and gap extension penalty 0.2. For multiple-sequence alignments (MSAs), Geneious Alignment with the same parameter, but Gap open penalty 20 and 5 refinement iterations was used. Pairwise identity within and between Metazoan/Apicomplexan proteins was calculated from MSAs and kernel smoothing was applied.

### Nucleosome reconstitution

DNA templates for nucleosome assembly were synthesized by PCR, using (Cy3-/Cy5-labelled) oligonucleotides binding to the flanking regions of the 147 bp NPS, derived from the nucleosome assembly 601 sequence, creating various linker lengths (Extended Data Fig. 2b): 0-NPS-0, 6-NPS-47, 77-NPS-77[47]. Analogously, plasmodial DNA templates were amplified from genomic DNA. Recombinant histones—canonical human and *P. falciparum* and variant *P. falciparum* (H2A.Z/B.Z containing)—were expressed, purified and refolded as previously described[8]. *Bos taurus* and *Gallus gallus* histone octamers were purified from calf thymus or chicken blood, respectively, as described previously[48,49] (Extended Data Fig. 2c). Nucleosome assembly was performed using the salt dialysis method. Histone octamers and DNA were mixed in 40–200 µl high-salt buffer (10 mM Tris pH 7.6, 2 M NaCl, 1 mM EDTA, 1 mM 2-mercaptoethanol, 0.05% Igepal CA-630) supplemented with 200 ng µl$^{-1}$ BSA in small dialysis chambers. High salt concentrations were reduced to 200 mM NaCl overnight at 4 °C. Reconstituted nucleosomes were analysed by gel electrophoresis on 6% polyacrylamide gels in 0.4× TBE buffer and visualized by fluorescence scanning (Typhoon, FLA-9500) or ethidium bromide staining (Extended Data Fig. 2d,g–l). Raw scans of the gels are provided in Supplementary Fig. 1.

### Protein expression and purification

*Pf*Snf2L (amino acids 250–1426) was sequence optimized for baculovirus mediated protein expression and 10×His-Tag purification. The coding sequence was cloned into the pFL plasmid and transformed into DH10BacEMYFP cells. The bacmid DNA was isolated and transfected into *Spodoptera frugiperda* Sf21 cells (Invitrogen) to produce initial virus and large-scale expression[50,51]. Cells expressing recombinant *Pf*Snf2L–10×His or *Hs*SNF2L–6×His were collected and lysed in 20 mM Tris–HCl pH 7.6, 500 mM KCl, 1.5 mM MgCl$_2$, 0.5 mM EGTA, 5 mM β-mercaptoethanol, 10% glycerol and 0.1% Igepal CA-630 using the Branson Sonifier 250. Purification was performed using NiNTA agarose (Qiagen) according to the manufacturer's recommendations. The protein concentration was estimated using the Bradford assay and the purity was checked using Coomassie-stained SDS–PAGE (Extended Data Fig. 2a). The enzymes Chd4, *Hs*SNF2L, *Pf*Snf2Lcore, *Hs*SNF2Lcore and *Pv*Snf2L (Supplementary Table 4) were generated and produced as described above.

### Nucleosome binding, assembly and remodelling assays

In vitro reconstituted nucleosomes in 20 mM Tris pH 7.6, 100 mM KCl, 1.5 mM MgCl$_2$, 0.5 mM EGTA, 10% glycerol, 200 ng µl$^{-1}$ BSA were incubated for 60 min at 30 °C with or without recombinant CREs. In competitive binding assays Cy3- and Cy5-labelled nucleosomes (15 nM) were incubated with increasing concentration of CREs. For non-fluorescent binding and assembly assays, 120 nM nucleosome and 700 nM CRE was used. Fluorescent remodelling assays contained 30 nM nucleosome, 1 mM ATP and variable concentrations of CREs. Assembly and remodelling reactions were stopped after 60 min (unless noted differently) by addition of 1 µg competitor plasmid DNA, and nucleosome positions were analysed by 6% native PAGE.

### ATPase assays

CRE (130–700 nM) in 20 mM Tris–HCl pH 7.6, 120 mM KCl, 1.5 mM MgCl$_2$, 0.5 mM EGTA and 10% glycerol was incubated with 500 µM ATP and 0.2 µCi $^{32}$P-γ-ATP in the absence or presence of mononucleosomes (60–350 nM) for 40 min at 30 °C. Released $^{32}$P-γ phosphate was separated from non-hydrolysed $^{32}$P-γ-ATP by thin-layer chromatography on PEI-Cellulose F plates (Merck, mobile phase: 50% acetic acid, 0.5 mM LiCl). After phosphoimaging (Typhoon FLA-9500), the signal intensities were quantified (Fuji Multi Gauge Software), the hydrolysis rate was calculated, corrected for chemical hydrolysis and normalized to CRE concentration.

### Plasmid construction

A synthetic gene comprising the native nucleotides 995–1894 of *PF3D7_1104200*, an artificial intron with a *loxP* site[52,53], nucleotides 1895–4278 recodonized and a 3×HA-tag-encoding sequence was ordered as a synthetic gene (IDT) and cloned into the vector pT2A-X-KO[54]. The resulting pT2A-Snf2L-KO plasmid contains a skip peptide sequence downstream of the *Pf*Snf2L-coding sequence, followed by a neomycin-resistance gene to enable selection-linked integration, a second *loxP* site, a *GFP* gene and an independent human DHFR gene.

### Parasite culture and transfection

*P. falciparum* clone 3D7 was cultured according to standard procedures in RPMI 1640 with AlbuMAX (Invitrogen), and synchronized as described previously[55]: schizonts were purified on a bed of 70% Percoll, incubated with new RBCs for 1–2 h, before leftover schizonts were removed with Percoll and subsequent sorbitol treatment. About 10 µg of plasmid was used for transfection of DiCre-expressing parasites using the Amaxa P3 primary cell 4D-Nucleofector X Kit L (Lonza)[35,56]. Successful transfection was selected with 2.5 nM of the antifolate WR99210 (Jacobus Pharmaceutical Company), starting 1 day after transfection. Resistant parasites were selected for genomic integration with 400 µg ml$^{-1}$ G418 (Sigma-Aldrich). After limiting dilution of drug-resistant parasites[57], genomic DNA of six clones was isolated using the Qiagen Blood and Tissue kit. Integration was confirmed by genotyping PCR using Q5 polymerase (NEB) and the primers listed in Supplementary Table 4. One stable clone was used for further phenotyping. Conditional DiCre-mediated recombination between *loxP* sites was performed as described previously[35]. For KO induction, synchronous parasites were treated with 100 nM rapamycin (Sigma-Aldrich) for 4 h (DMSO treatment as control), washed and returned to culture. Gene excision was confirmed by genotyping PCR, mRNA levels were checked in RNA-seq samples and protein levels were analysed using western blotting.

### Western blot analysis

Parasites were isolated from RBCs by 0.1% saponin and protease inhibitor cocktail PIC (Roche) in PBS and boiled for 10 min in 62 mM Tris pH 6.5, 25% glycerol, 2% SDS, 0.2 M DTT, 0.05% OrangeG. Proteins were separated on 4–20% SDS gels (BioRad), and transferred onto a nitrocellulose membrane before immunoblotting. In this study, we used rat anti-HA (1:2,000, Roche) and mouse anti-Enolase (1:1,000, G. K. Jarori) antibodies, conjugated secondary antibodies and Odyssey imaging system (LiCOR Biosciences) according to manufacturer's recommendation.

### IFA

Immunofluorescence assays were performed as described previously[56] with the addition of 0.0075% glutaraldehyde during fixation. Primary antibodies (rat anti-HA, rabbit anti-Gap45[58], rabbit anti-Ama1[59]) were used at a dilution of 1:500. For image acquisition, the Leica DMi8 Widefield microscope was used; image processing and quantification were performed using Fiji v.2.9.0 (ImageJ).

## Analysis of parasite growth, reproduction and egress

Growth was determined by microscopy counting of parasites from Giemsa-stained thin blood films and expressed as percentage parasitaemia (percentage of infected RBCs/RBCs). For reproduction and egress assays, equal numbers of mature schizonts were Percoll-purified. For parasite reproduction, parasites were incubated in 4 ml RPMI with 1% haematocrit for 2 h, before the number of rings was quantified from Giemsa stain. Live imaging of parasite egress was performed as described previously[56], and parasites were binned into groups 15 s after egress (normal/clustered merozoites). For RBC staining, RPMI was supplemented with 0.1× Phalloidin 594 Conjugate (Abcam) during live imaging. The RBC ghost location 40 s after egress was categorized (distant/attached/overlapping).

## TEM analysis

Schizont-stage parasites (46 h.p.i.) were treated with compound 2 (ref. 60) for 2 h for further maturation. The parasites were then washed in PHEM buffer (2.5 mM $MgCl_2$, 35 mM KCl, 5 mM EGTA, 10 mM HEPES, 30 mM PIPES, pH 7.2), fixed for 1 h in a solution containing 2.5% glutaraldehyde, 4% formaldehyde, 4% sucrose in 0.1 M PHEM buffer, post-fixed in 1% $OsO_4$ plus 0.8% ferrocyanide and 5 mM $CaCl_2$ in 0.1 M cacodylate buffer for 1 h, washed twice in cacodylate buffer (10 min, 1 h) and double-distilled water (5 min, 15 min), respectively, and dehydrated in a graded acetone series which included en bloc staining with 1% uranyl acetate in the 20% step. The cells were finally embedded in Epon812 epoxy resin to enable ultrathin sectioning. To carry out electron microscopy, the ultrathin sections were post-stained with 1% lead citrate for 2 min. Transmission electron microscopy (TEM) was performed using the JEOL F200 (JEOL) system operated at 200 kV equipped with a XAROSA 20 mega pixel CMOS camera (EMSIS).

## Preparation of nuclear extracts and co-IP experiments

Nuclei of PfSnf2L–HA and 3D7 parasites were prepared as described previously[61] and treated with 0.5 U μl⁻¹ benzonase (Sigma-Aldrich) in 20 mM HEPES pH 7.9, 10 mM KCl, 0.1 mM EDTA, 0.1 mM EGTA, 1 mM DTT, 0.65% Igepal CA-630 and PIC for 30 min at room temperature. For extraction of nuclear proteins, KCl was added to 0.4 M, incubated for 30 min at room temperature and the insoluble fraction was removed by centrifugation (5,000$g$, 30 min, 4 °C). The supernatant was diluted with 2.5 vol of 50 mM Tris pH 7.4, 50 mM NaCl, 1 mM EDTA, 1% Igepal CA-630 and PIC, and incubated overnight at 4 °C under constant agitation with equilibrated anti-HA magnetic beads (Thermo Fisher Scientific). Beads were pelleted using a magnetic rack, washed three times with latter buffer and three times with 50 mM Tris-HCl pH 8. The beads were dried, stored at −20 °C and used for LC−MS/MS analysis of co-immunoprecipitated proteins in PfSnf2L–HA parasites (3D7 as a negative control), in triplicates each. The collected fractions (input, flowthrough, beads) were analysed on 4–20% SDS−PAGE, silver-stained and, in an additional version, were probed with anti-HA antibodies on a western blot.

## LC−MS/MS and data analysis

Beads were incubated with 10 ng μl⁻¹ trypsin in 1 M urea and 50 mM $NH_4HCO_3$ for 30 min, washed with 50 mM $NH_4HCO_3$ and the supernatant was digested overnight in presence of 1 mM DTT. Digested peptides were alkylated and desalted before LC−MS analysis. For LC−MS/MS purposes, desalted peptides were injected into the Ultimate 3000 RSLC-nano system (Thermo Fisher Scientific), separated in a 15 cm analytical column (75 μm inner diameter with ReproSil-Pur C18-AQ 2.4 μm from Dr Maisch) with a 50 min gradient from 4% to 40% acetonitrile in 0.1% formic acid. The effluent from the HPLC was directly electrosprayed into the Q Exactive HF system (Thermo Fisher Scientific) operated in data-dependent mode to automatically switch between full scan MS and MS/MS acquisition. Survey full scan MS spectra (from $m/z$ 350

to 1,600) were acquired with resolution $R = 60,000$ at $m/z$ 400 (AGC target of $3 × 10^6$). The 10 most intense peptide ions with charge states between 2 and 5 were sequentially isolated to a target value of $1 × 10^5$ and fragmented at 30% normalized collision energy. Typical mass spectrometric conditions were as follows: spray voltage, 1.5 kV; heated capillary temperature, 275 °C; ion-selection threshold, 33.000 counts.

MaxQuant v.1.6.14.0 was used to identify proteins and quantification was performed using iBAQ with the following parameters: database Uniprot_UP000001450_Plasmodiumfalciparum_20201007.fasta; MS tol, 10 ppm; MS/MS tol, 20 ppm Da; PSM FDR, 0.01; protein FDR, 0.01 min; peptide length, 7; variable modifications, oxidation (M), acetyl (protein N-term); fixed modifications, carbamidomethyl (C); peptides for protein quantitation, razor and unique; min. peptides, 1; min. ratio count, 2. MaxQuant iBAQ values were $\log_2$-transformed, and missing values imputed with 8. Ribosomal proteins and hits detected with only 1 peptide were excluded. Identified proteins were considered as interaction partners of the bait, if $\log_2$[Snf2L-3HA] − $\log_2$(3D7) > 3 or $\log_2$[Snf2L–3HA] − $\log_2$(3D7) > 2 and FDR < 0.05. The MS proteomics data have been deposited at the ProteomeXchange Consortium via the PRIDE[62] partner repository under the dataset identifier PXD041155.

## RNA-seq and data analysis

RNA-seq analysis with or without *Snf2L* KO and with or without drug, respectively, was performed in triplicates. Total RNA from infected RBCs containing highly synchronized parasites was isolated using the Whole Blood Quick RNA kit (Zymo Research) according to the manufacturer's protocol. The RNA quality was checked using 4200 TapeStation System (Agilent) and 300 ng was used as the input for Illumina Stranded mRNA Prep Ligation (Illumina). Libraries were sequenced on the Illumina NextSeq 2000 sequencing system. Sequenced reads (2 × 57 bp, paired-end, ~20 million reads per sample) were trimmed using trimmomatic (v.0.39)[63] and mapped to the *P. falciparum* 3D7 genome v3.0 (https://PlasmoDB.org, release 52)[64] using STAR (v.2.7.9a)[65]. PlasmoDB annotation was converted to GTF-format using gffread (v.0.12.1)[66]. Preprocessing and mapping quality control was done using FastQC (v.0.11.8)[67], qualimap (v.2.2.2d)[68], samtools (v.1.12)[69] and multiqc (v.1.9)[70]. The pipeline was implemented with snakemake (v.5.32.0)[71] and is available at GitHub (https://github.com/SimHolz/Watzlowik_et_al_2023), in addition to the R scripts. RSubread/FeatureCounts (v.2.12.2) was used to calculate read counts, while differential expression analysis was performed using DESeq2[72,73], with adjusted $P < 0.05$ used as the significance cut-off. Further analysis and visualization were done in R[74] using tidyverse[75] and ggpubr[76]. The degPatterns function from DEGreport[77] was used for clustering. For expression quantile calculation, transcripts per millions normalization was used. RNA-seq data were submitted to the Gene Expression Omnibus (GEO) database (GSE228949).

RNA-seq-based cell cycle progression was estimated in R by comparing the normalized expression values of each sample to the RNA-seq data from a previous study[78] using a statistical model previously described[79].

Pf*Snf2L*-KO efficacy was estimated by mapping reads to the sequence of the recodonized transfected *Snf2L* gene and counting reads mapped to the $Snf2L_{recodon}$ part of the gene, which is disintegrated after KO induction.

## MNase-seq and data analysis

MNase-digestion was adapted from a previous study[3] with the following modifications: highly synchronous parasites were cross-linked and stopped as described. RBCs were lysed and nuclei were isolated as described for pull-down experiments. Nuclei were resuspended in 75 μl 50 mM Tris pH 7.4, 4 mM $MgCl_2$, 1 mM $CaCl_2$, 0.075% NP40, 1 mM DTT, PIC with 0.75 U Micrococcal nuclease (Worthington Biochemicals) and 50 U exonuclease III (NEB). Each sample was aliquoted in 3 × 25 μl and incubated for 2.5, 5 and 10 min at 37 °C under agitation (low, mid

and high digestion), before the reaction was stopped by adding 25 µl 2% Triton X-100, 0.6% SDS, 300 mM NaCl, 6 mM EDTA/PIC and placed at 4 °C. De-cross-linking was performed at 45 °C overnight after adjusting to 1% SDS, 0.1 M NaHCO$_3$ and 0.5 M NaCl. Proteins were digested by addition of 40 µg of proteinase K (Zymo Research) and incubation for 1 h at 55 °C. Subsequently, DNA was isolated using the EXTRACTME DNA clean-up kit and Micro Spin columns (Blirt) according to the manufacturer's recommends. Sequencing and data analysis is described in Supplementary Note 1.

### ADP biosensor assay

ATPase activity and inhibition were measured using the ADP biosensor assay: in 20 µl, 0.2 µM TMR-maleimide-labelled ParM (prepared as described previously[80]) was mixed with 125 µM ATP, 100 ng plasmid pT11 DNA, 5 µM H4 peptide (amino acids 8–25, AnaSpec) in 10 mM Tris pH 8.6, 1.5 mM MgCl$_2$, 100 mM KCl, 0.01% pluronic in presence (or absence as a negative control) of 0.4 µM recombinantly expressed *Pf*Snf2L and NH125 at varying concentrations. The resulting ADP binding to ParM is expressed by increasing fluorescence intensity and was kinetically measured at 28 °C in 2 min intervals over 2 h in the Tecan infinite F500 reader. The signal was normalized to timepoint 0.

### Toxicity assay for *Plasmodium*, *Toxoplasma* and HeLa

Toxicity for *Plasmodium* was determined by culturing in presence of NH125 and DMSO for 72 h and subsequent quantification of parasitaemia in Giemsa-stained blood films. Toxicity for *Toxoplasma gondii* was tested using a plaque assay, whereby 1,000 *T. gondii* tachyzoites inoculated onto human foreskin fibroblasts as described previously[81] were treated with NH125 and DMSO, fixed and stained 6 days later and evaluated for plaque formation. Toxicity on human HeLa cells, cultured as described previously[49], was investigated by 48 h NH125/DMSO treatment and subsequent monitoring of metabolic activity using the Cell Proliferation Kit II-XTT (Sigma-Aldrich) according to the manufacturer's recommendations. Half-maximal effective concentration (EC$_{50}$) values were obtained by fitting the dose–response model using three-parameter log-logistic models and estimated for *T. gondii*.

### Gametocyte induction

Sexual commitment was induced using the nutrient deprivation induction method as described previously[82]. In brief, parasites were tightly synchronized using Percoll (63% isotonic solution) density-gradient centrifugation to isolate mature-stage schizonts and allow invasion of naive RBCs, followed by treatment with 5% sorbitol to kill the remaining schizonts in the culture and retain only young rings (day 0). The assay was performed in six-well plates and started at 1.5% parasitaemia and 3% haematocrit. For a high-throughput sexual-commitment–conversion assay in a 96-well format, parasites were plated at 0.5% parasitaemia and 2.5% haematocrit. Drug treatment with NH125 and/or rapamycin treatment for the inducible KO line were started at 10 h.p.i. in complete medium supplemented with choline chloride as indicated in Extended Data Fig. 9a. To induce sexual commitment, parasites were shifted to minimal fatty acid (mFA) medium at 20–24 h.p.i. (day 1). mFA medium was prepared by supplementing incomplete medium (RPMI-1640 without AlbuMAX) with 0.39% fatty-acid-free BSA (Sigma-Aldrich), 30 µM oleic acid (Sigma-Aldrich) and 30 µM palmitic acid (Sigma-Aldrich). Then, 22–26 h after induction (day 2), the mFA medium was replaced with complete RPMI medium (with AlbuMAX). At this timepoint, in the high-throughput assay, the parasites were treated with 50 µM E64 to block merozoite egress and measure the sexual commitment rate based on the number of AP2–GGFP-positive schizonts. On day 3, parasitaemia was quantified using Giemsa-stained blood smears or by flow cytometry using Hoechst staining, and 20 U ml$^{-1}$ heparin was added to the culture until day 6. The medium was changed daily until quantification of gametocytemia on day 9 by Giemsa staining to calculate sexual conversion rate as follows: sexual conversion rate = gametocytemia on day

9/parasitaemia on day 3. Within the sexual-commitment-conversion assay, parasite conversion was analysed with flow cytometry on day 6 after staining with TubulinTracker Deep Red and Hoechst, or on day 7 based on GEXP-02 positive cells (Extended Data Fig. 10).

### Dose–response assays

Half-maximal inhibitory concentrations (IC$_{50}$ values) for NH125 were determined against various strains by exposing ring-stage parasites to a range of drug concentrations (twofold serial dilutions starting from 5 µM) in 96-well plates. Assays were seeded at 0.2% starting parasitaemia and 1% haematocrit. Parasite growth in each well was determined after 72 h by flow cytometry after staining with Mito Tracker Deep Red and Hoechst. Assays were performed in duplicate. IC$_{50}$ values were calculated from three to four independent replicates by nonlinear regression analysis in GraphPad Prism.

### ChIP–seq and data analysis

ChIP experiments were performed on *Pf*Snf2L–HA parasites at 10, 22, 34 and 46 h.p.i. in duplicate with an additional no-epitope control on WT 3D7 parasites, respectively. Highly synchronous parasites were cross-linked and stopped as described previously[3]. RBCs were lysed and nuclei were isolated as for co-IP experiments. Nucleus isolation was followed by chromatin sonication using the Covaris Focus-Ultrasonicator (5% duty cycle, 75 W peak incident power, 200 cycles per burst, 7 °C, 5 min), protein–chromatin complex immunoprecipitation using anti-HA antibodies and DNA purification (Qiagen MinElute Kit) as performed previously[83,84]. All ChIP libraries (with paired non-immunoprecipitated input control samples) were prepared as described previously[83,84], and checked for library sample quality (high-sensitivity DNA Qubit Fluorometer) and sequence length (Agilent TapeStation 4150) before sequencing on the Illumina NextSeq 2000 system for P1 150 × 150 paired-end sequencing. ChIP reads were first trimmed using Trimmomatic v.0.32.3 (<30 phred, SLIDINGWINDOW:4:30 option)[63]. The pre- and post-trimming read quality was assessed using FastQC (v.0.11.9)[67]. Filtered and trimmed reads were then mapped using BWA-MEM (v.0.7.17.2)[85] using paired-end simple Illumina mode to the *P. falciparum* 3D7 genome (https://PlasmoDB.org; release 52) and filtered for multi-mapped reads (MAPQ = 1 option). Enrichment was calculated as log$_2$[ChIP/input] using deepTools bamCompare (v3.5.2)[86], averaged for replicates and aligned to +1 nucleosome as for nucleosome occupancy. ChIP–seq data were submitted to the GEO (GSE237217).

### Reporting summary

Further information on research design is available in the Nature Portfolio Reporting Summary linked to this article.

## Data availability

RNA-seq and MNase-seq data generated for this study have been deposited in the GEO (GSE228949), as were ChIP–seq data (GSE237217). Proteomic data have been deposited at ProteomeXchange (PXD041155).

## Code availability

The sequencing analysis pipeline and code to reproduce the analysis is available at GitHub (https://github.com/SimHolz/Watzlowik_et_al_2023).

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

**Acknowledgements** We thank the staff at Fraunhofer ITEM Regensburg and 2bind for their contribution in the inhibitor screening process; the members of the Protein Analysis Unit (ZfP, LMU Munich) for performing MS; the staff at the Genomics Core Unit (KFB, University of Regensburg) for their sequencing service; T. Spielmann for proving DNA constructs; and M. Blackman for the gift of antibodies. Research was funded by a DFG Programme Grant (project number: 534335380) to G.L. and M. Meissner and by a DFG Equipment grant (INST 86/1831-1). R.S. and V.A.B. were supported by National Institutes of Health NIH/NIGMS T32 GM125592-01. A.K. was supported by a DFG equipment grant (INST 86/1852-1). K.V., B.S., L.S. and M. Marti are supported by Wellcome Trust Investigator award 223210/Z/21/Z.

**Author contributions** G.L., M.T.W., M. Meissner, M.L. and M. Marti conceived and designed experiments with input of S.D. M.T.W. performed most of the experiments. S.D. established transgenic parasites and imaging methods. S.H. developed MNase-seq analysis pipeline and supported library preparation. Sequencing data were analysed by S.H. and M.T.W. E. Silberhorn produced recombinant proteins and performed in vitro assays. A.K. performed electron microscopy. M.G. performed *T. gondii* toxicity assays. V.A.B. performed ChIP–seq. R.S. performed gametocyte induction assays. B.S. performed toxicity assays. K.V. established the dual reporter line and performed the sexual commitment conversion assays with E. Schadt. L.S. generated AP2-G KO parasites. M.T.W., M. Meissner. and G.L. wrote the manuscript. All of the authors discussed and approved the manuscript.

**Funding** Open access funding provided by Universität Regensburg.

**Competing interests** University of Regensburg has registered a patent for NH125 as CRE inhibitor with G.L., M.T.W. and 2bind listed as inventors (application number EP 23 219 442.3). The other authors declare no competing interests.

**Additional information**

**Correspondence and requests for materials** should be addressed to Markus Meissner or Gernot Längst.

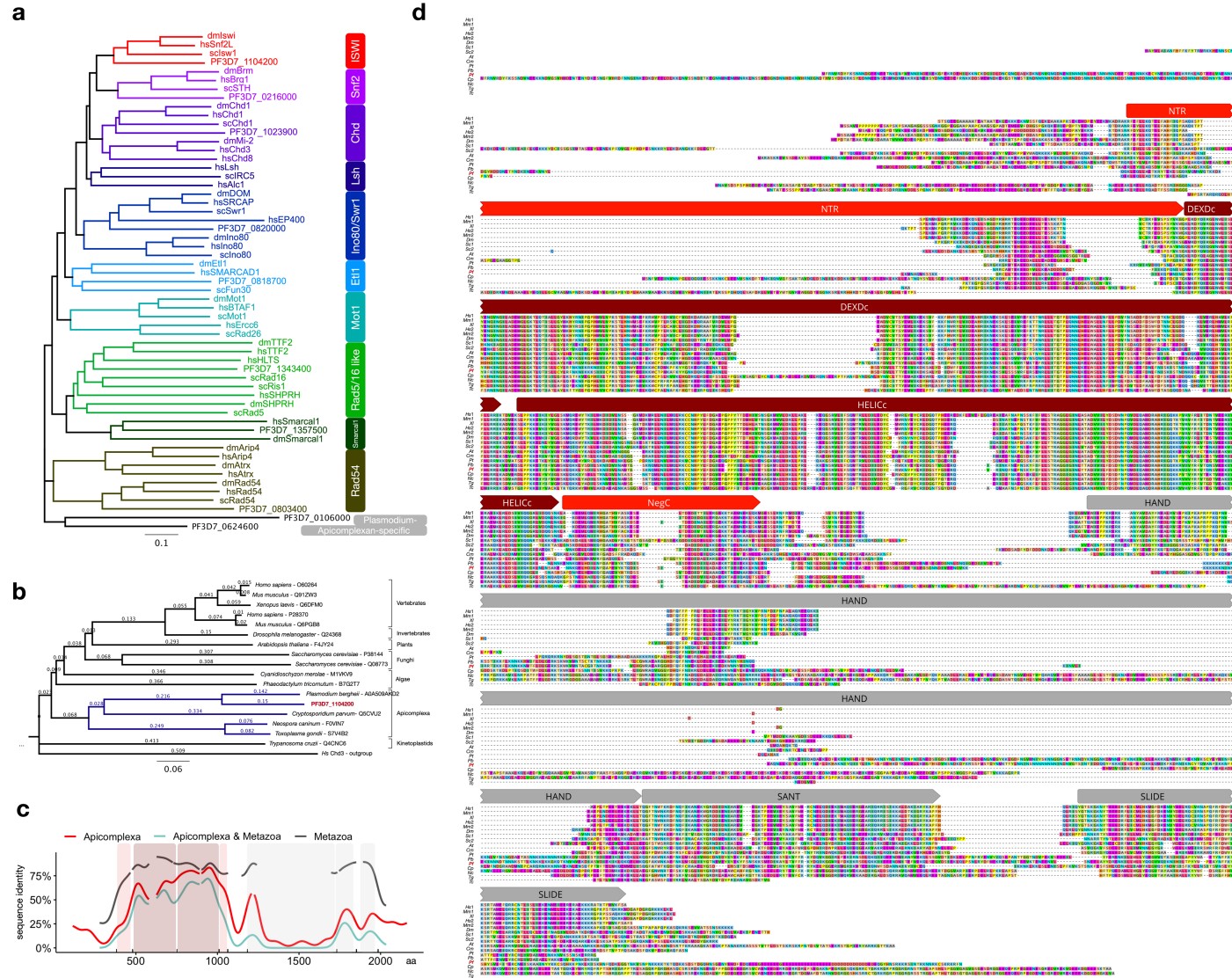

**Extended Data Fig. 1 | *Pf*Snf2L is an ISWI-related chromatin remodelling enzyme with high sequence divergence.** Protein sequences used for in silico analysis are listed in Supplementary Table 3. **a**, Phylogenetic analysis of the *Pf*Snf2L (Pf3D7_1104200) helicase region with subfamily representatives of *Homo sapiens (Hs)*, *Drosophila melanogaster (Dm)*, *Saccharomyces cerevisiae (Sc)* and predicted *Pf* SWI2/SNF2-ATPases. Enzymes of the same Snf2 helicase subgroups according to[1], are shown in identical colours. **b**, Rooted phylogenetic tree of the full length ISWI homologues from different organisms, representing selected eukaryotic kingdoms with Apicomplexa being highlighted in blue and

*Hs*Chd3, included as an non ISWI type enzyme (outgroup). Numbers indicate the substitutions per site. **c**, Pairwise identity of ISWI enzymes within and between Metazoa, and Apicomplexa based on the multiple sequence alignment in d. Gaps represent insertions and the colouring shows the predicted domains as given in Fig. 1a. **d**, Multiple sequence alignment of ISWI homologues from *Hs, Mm, Xl, Dm, Sc, At, Cm, Pt, Pb, Cp, Nc, Tg, Tc* and *Pf* (*Pf*Snf2L, PF3D7_1104200) including all predicted domains as shown in Fig. 1a (the colour code is defined by the amino acids).

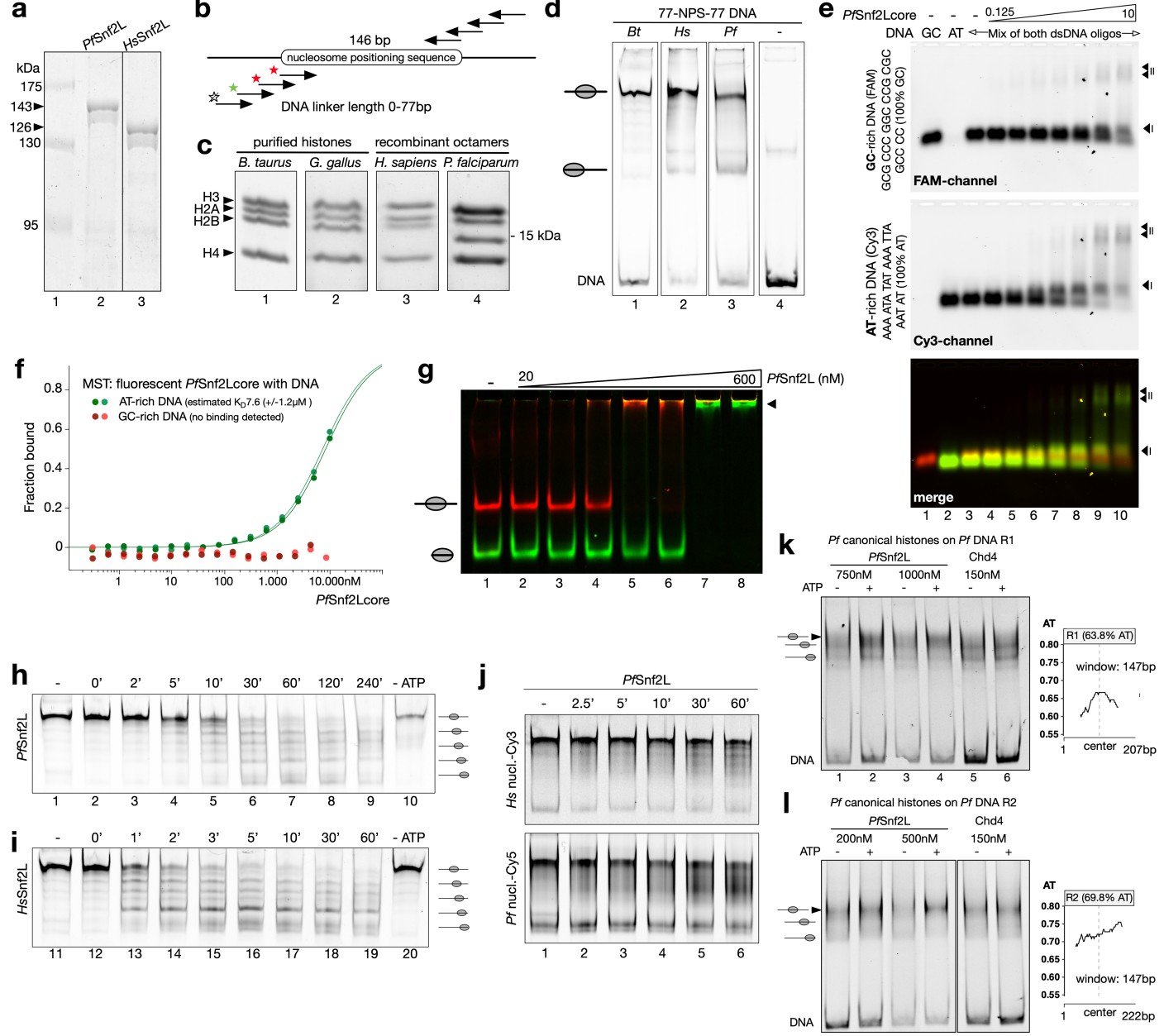

**Extended Data Fig. 2** | See next page for caption.

**Extended Data Fig. 2 | Recombinant *Pf*Snf2L binds preferentially binding to AT-rich nucleosomal DNA and exerts ATP-dependent remodelling with reconstituted plasmodium nucleosomes. a**, SDS-PAGE (6%, Coomassie-stained) of recombinantly expressed and purified remodelling enzymes (lanes 2 and 3). Protein band sizes are indicated, and the protein marker is shown (lane 1). One representative gel of n = 4 independent protein expression replicates is shown. **b**, Schematic representation of DNA template generation by PCR for nucleosome assembly. The DNA template contains a nucleosome positioning sequence (NPS) with different linker lengths and is differentially fluorescently labelled with the primers marked by a star. **c**, SDS-PAGE (15%, Coomassie-stained) of purified (*Bt* and *Gg*) or recombinantly expressed (*Hs* and *Pf*) and purified histone octamer preparations. Representative images of n = 4 (*Bt*)/n = 3 (*Gg*, *Hs*, *Pf*), independent histone purification replicates are shown. Histone identities are indicated, and the size of the protein marker is shown. **d**, Native PAGE with reconstituted nucleosomes on Cy5-labelled 77-NPS-77 DNA (lane 4) using the octamers species indicated above the lanes. Relative migration positions of nucleosomes and free DNA are marked. One representative gel of n = 3 independent nucleosome reconstitution replicates is shown. **e**, Competitive binding assay using *Pf*Snf2Lcore enzyme (0.125–10 µM) and an equimolar mixture of GC-DNA-FAM (upper panel, red in merge) and AT-DNA-Cy3 (mid panel, green in merge). The fast (I) and slow migrating (II) shift products are marked by triangles. The DNA sequences are indicated above the EMSA. **f**, MST analysis using the fluorescently labelled *Pf*Snf2Lcore protein (His-binder, NanoTemper) with increasing concentrations of the non-labelled GC-rich (dark red and light red dots represent the replicates) and AT-rich DNA (dark green and light green dots represent the replicates). Binding curves are fitted where possible, and an estimated $K_D$ value is shown. **g**, Competitive nucleosome binding assay using *Pf*Snf2L (20–600 nM) and an equimolar mixture of *Gg*/6-NPS-47-Cy5 (red) and *Gg*/0-NPS-0-Cy3 (green) nucleosomes. Merged channels are shown with nucleosome positions and shift products (triangle) indicated. **h**, Nucleosome remodelling assay using 300 nM *Pf*Snf2L and centre-positioned nucleosomes (*Bt*/77-NPS-77-Cy5), incubated for increasing reaction time (0 to 240 min, as indicated). The relative positions of nucleosomes are indicated. One representative gel of n = 2 independent replicates is shown. **i**, Nucleosome remodelling assay using 300 nM *Hs*Snf2L as described in h. **j**, Competitive remodelling assay using *Pf*Snf2L (300 nM) and an equimolar mixture of nucleosomes reconstituted with human (*Hs*) or Plasmodium (*Pf*) histone octamers, on Cy3 or Cy5 labelled 77-NPS-77 DNA. The nucleosome mixture was remodelled for the indicated time, the reaction was stopped, and nucleosome positioning was analysed by native PAGE and differential fluorescence imaging. Lane 1 serves as a control, lacking ATP in the reaction. One representative gel of n = 6 independent replicates is shown. **k**, Nucleosome remodelling assay using *Pf*Snf2L (750/1000 nM) and *Hs*Chd4 (150 nM) as control, using nucleosomes reconstituted with canonical *Pf* histone octamers on a genomic plasmodium sequence with an average of 63.8% AT content (R1-DNA). The central, nucleosome binding region of the DNA exhibits an average AT content of about 67%. The plot (on the right) shows the average AT content for a window size of 147 bp. Nucleosome positioning is analysed on 6% native PAA gels, stained with ethidium bromide. The relative positions of the nucleosomes are indicated on the left. **l**, Nucleosome remodelling assay as described in k, using *Pf*Snf2L (500/750 nM) and *Hs*Chd4 on a plasmodium genomic sequence (R2-DNA; average AT content of 69.8%) reconstituted into nucleosomes. The central, nucleosome binding region of the DNA template exhibits an average AT content of 73%, as shown in the plot on the right.

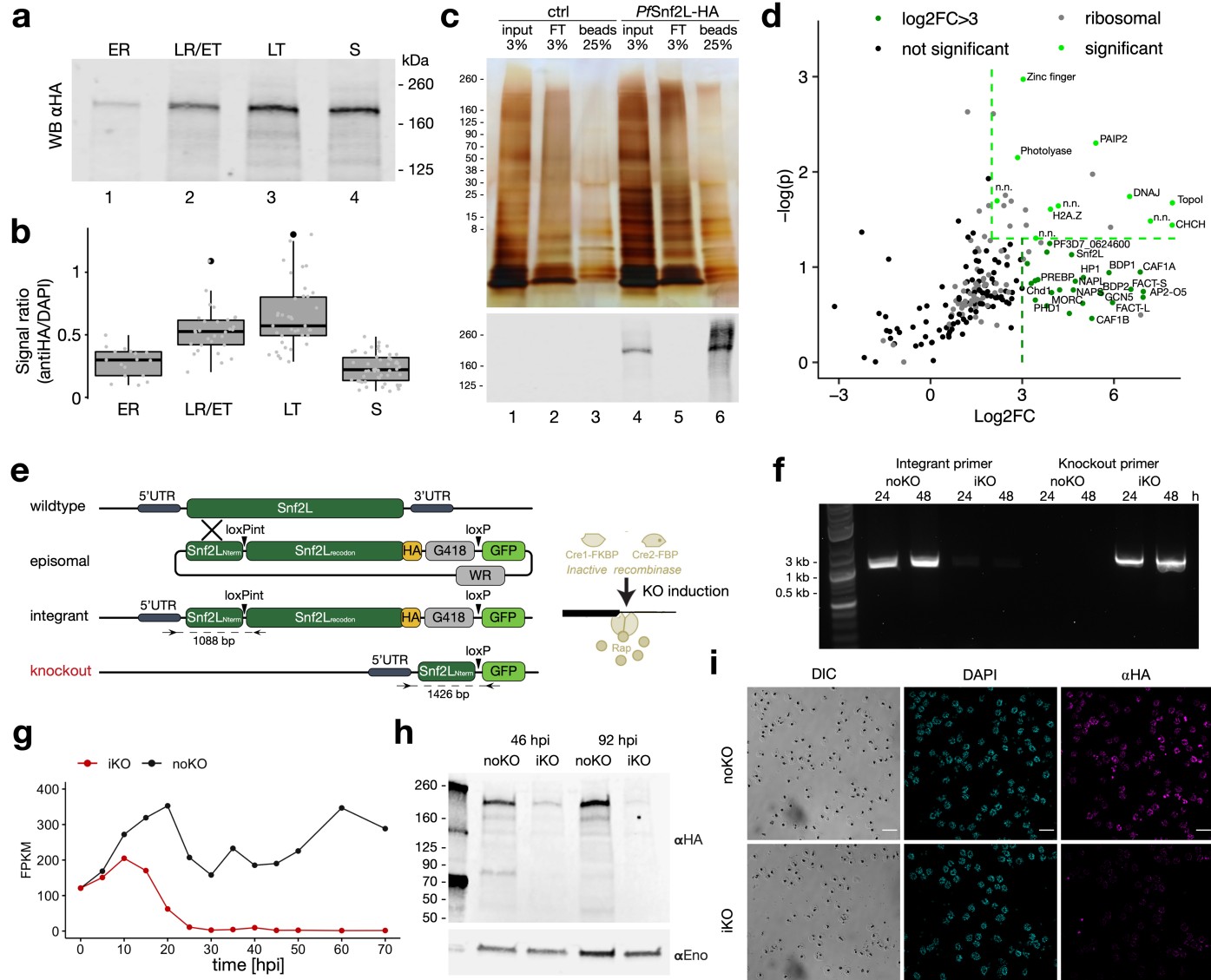

**Extended Data Fig. 3 | Endogenous tagged *Pf*Snf2L in parasites reveals its nuclear localization, allows the identification of *Pf*Snf2L interactors and is designed as a conditional knockout system. a**, Western blot analysis using an antiHA antibody on whole cell extract of synchronized *Pf*Snf2L-HA expressing parasites harvested after 11 (ER: early ring), 22 (LR/ET: late ring/early trophozoite), 33 (LT: late trophozoite) and 44 h.p.i. (S: schizont). Equal number of iRBCs are loaded on 4–20 % SDS-PAGE and analysed by Western blot. One representative gel of n = 2 independent replicates is shown. The position of the protein markers and the sizes are shown on the right. **b**, Boxplots showing the quantification of the IFA given in Fig. 1e. For n = 178 parasites of n = 1 biological replicate in four different stages, antiHA and DAPI signal intensities were measured, corrected for background and the antiHA/DAPI signal ratio for single parasites is plotted. The central line of the box represents the median, the box spans the interquartile range (IQR), and the whiskers extend to 1.5 times the IQR. Dots represent outliers beyond this range. **c**, SDS-PAGE (4–20%) of antiHA-IP samples - input, flowthrough (FT) and beads of control and PfSnf2L-HA cells are analysed, silver-stained (upper panel) and probed with an antiHA antibody on western blot (lower panel). One representative gel of n = 3 independent replicates is shown. The sizes of the protein markers are shown on the left. **d**, Vulcano plot of *Pf*Snf2L-IP LC-MS/MS performed as triplicates. The results are showing the -log(p-value) of a two-sided t.test over the log2FC(IP/ctrl). Proteins are indicated with short names or *n.n.* where the function is unknown. **e**, Schematic representation of the strategy to generate transgenic *Pf*Snf2L-HA-cKO parasites. The wildtype locus, the episomal plasmid, the locus after endogenous integration and the locus after excision through rapamycin-activated DiCre recombinase (right), as well as the primers used for genotyping PCR are described. Abbreviations. Rap: rapamycin, UTR: untranslated region, Nterm: N-terminal, recodon: recodonized, loxPint: artificial intron with loxP site, G418/WR: resistance genes, GFP: green-fluorescent protein. **f**, Validation of *Pf*Snf2L-knockout on DNA level by genotyping PCR 24/48 h post KO induction with primers indicated in e. n = 1 replicate was performed. **g**, Determination of the KO efficacy of *Pf*Snf2L on mRNA level in synchronized parasites. The quantification of the RNA-seq data at the given time point after infection (h.p.i.) is shown. One representative experiment of n = 3 replicates is shown. **h**, Western blot analysis of *Pf*Snf2L levels with antiHA antibody in whole cell extracts with (iKO) and without (noKO) rapamycin one (46 h.p.i.) and two (92 h.p.i.) cycles after induction. Analysis with the antiEnolase antibody served as loading control. One representative gel of n = 2 independent replicates is shown. The non-specific staining of the protein marker is shown in lane 1. **i**, Immunofluorescence analysis of schizont stage parasites one cycle after KO induction with antiHA antibody and DAPI (scale bar: 20 µm). One representative image of n = 3 independent replicates is shown.

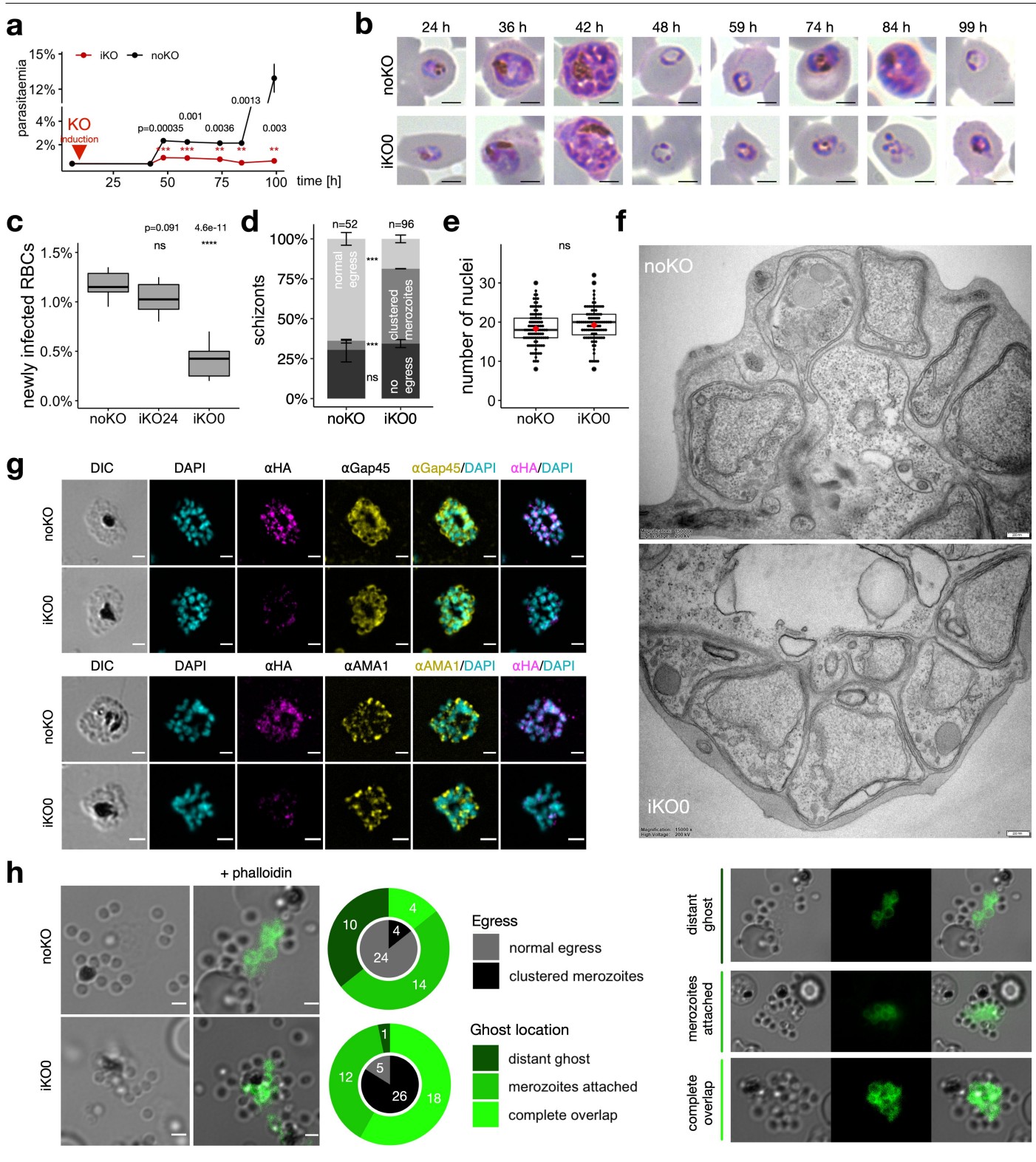

**Extended Data Fig. 4** | See next page for caption.

**Extended Data Fig. 4 | *Pf*Snf2L-KO parasites show developmental abnormalities and a red blood cell egress defect. a**, Growth curve analysis of synchronized noKO and iKO (0 h.p.i.) parasites as a biological replicate of Fig. 2a. Mean ± SD of n = 4 technical replicates (with n = 100 parasites for each time point) and the significances are shown. **b**, Representative Giemsa-stained blood smears of parasites treated as in a, scale bar: 2 µm. **c**, Reproduction analysis of noKO/iKO24 (24 h.p.i.)/iKO0 (0 h.p.i.) parasites (number of parasites n > 100 in each group with n = 1 biological replicate). The central line of the box represents the median, the box spans the interquartile range (IQR), and the whiskers extend to 1.5 times the IQR. **d**, Egress classification and quantification (Mean ± SD) of noKO/iKO0 (0 h.p.i.) parasites, using 12 independent live imaging videos with n = 2 biological replicates. The number of analysed parasites (n) is shown. **e**, Quantification of nuclei 46 h.p.i. of noKO and iKO0 (0 h.p.i.) parasites (number of parasite n = 100 in each group within n = 1 biological replicate). The central line of the box represents the median, the box spans the interquartile range (IQR), and the whiskers extend to 1.5 times the IQR. Dots represent individual parasites; large dots are outliers beyond the described range. **f**, Representative TEM image of mature noKO and iKO0 (0 h.p.i.) schizonts. Scale bar: 0.2 µm. One representative image of n = 2 independent replicates is shown. **g**, Immunofluorescence analysis of noKO/iKO0 (0 h.p.i.) schizonts with DAPI, antiHA and antiGap45 as cytogenesis markers, and antiAMA1 as apicoplast marker. Scale bar: 2 µm, One representative image of n = 3 independent replicates is shown. **h**, Representative images, egress quantification and RBC ghost location during phalloidin-supplemented live imaging of individual egressed noKO/iKO schizonts in 6 independent live imaging videos within n = 1 biological replicates. Number of parasites n is shown. Exemplary images of ghost location classification are shown on the right. Scale bar: 1 µm. a,c-e, two-tailed Student's t-test, unpaired, ns p > 0.05, * p < 0.05, ** p < 0.01, ***p < 0.005.

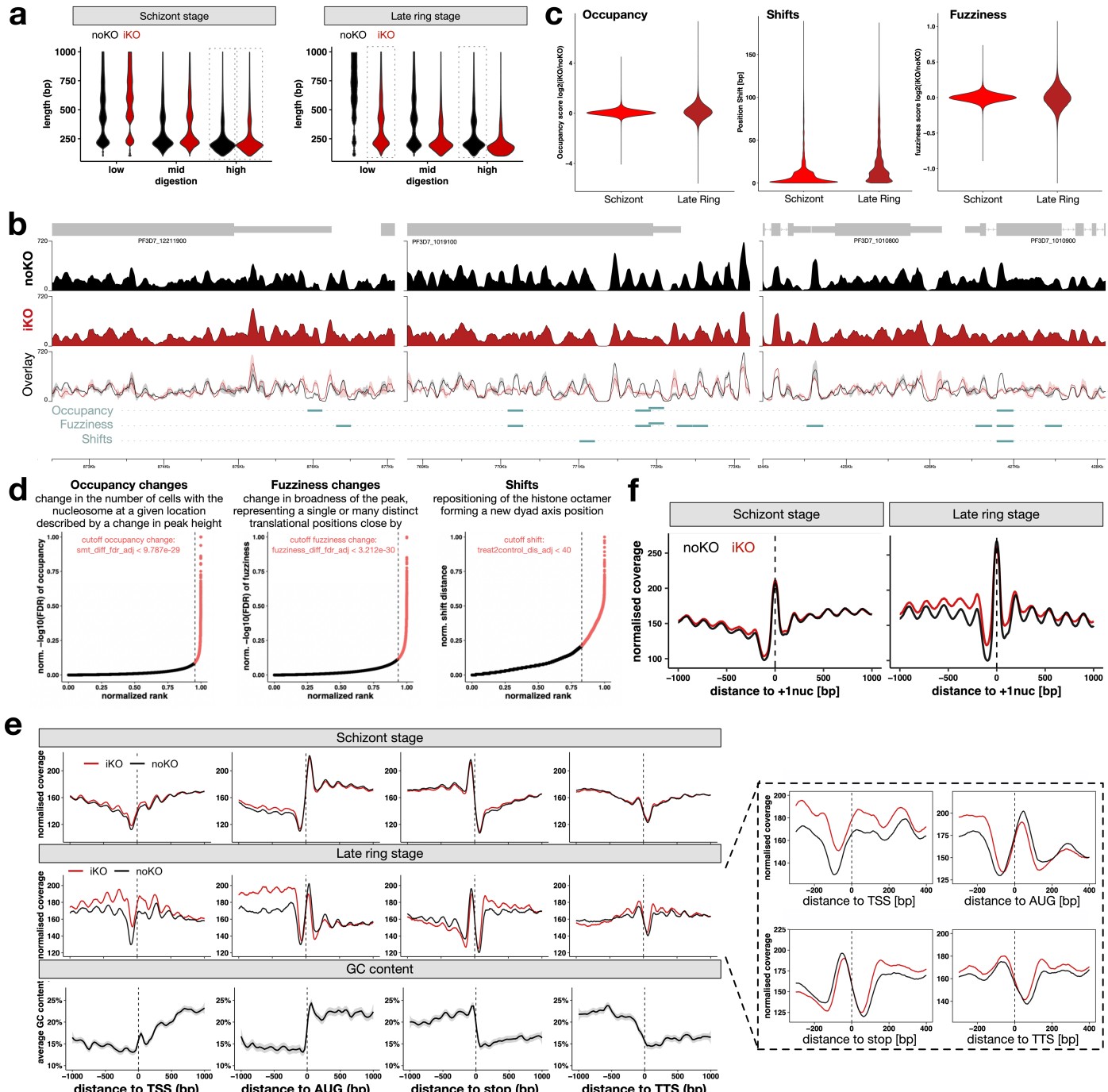

**Extended Data Fig. 5 | MNase-Seq analysis of *Pf*Snf2L-KO parasites exhibit reduced chromatin stability and altered nucleosome positions in late ring stage. a**, Normalized DNA size distribution of MNase-digested chromatin at schizont/late ring stage of noKO/iKO parasites. One representative replicate of n = 3 is shown, dotted boxes indicate samples used for high throughput sequencing analysis. **b**, Exemplary coverage plots of MNase-seq fragments of late ring stage parasites upon PfSnf2L-KO. The mean of duplicate experiments is shown. The range between minimum and maximum values of the replicates is indicated by a shaded area in the overlay. Nucleosome positions with identified changes in occupancy, fuzziness and shifts are plotted below. Regions shown are chr12:872500–877200, chr10:768800–773200, chr10:424200–428200 with annotated genes marked in grey. **c**, Violin plots showing Log2 fold changes

(iKO/noKO) of nucleosome occupancy scores (left), fuzziness scores (right) and position shifts (middle) in the schizont and late ring stages. Significant differences in variances are observed (Levene's Test, p < 0.001 for all comparisons). **d**, Definition of nucleosome positioning changes and their ranked normalized dynamics parameter from DANPOS analysis of late ring stage iKO/noKO chromatin. The cutoff is defined as slope=1 and shown as a dotted line. **e**, MNase-seq occupancy profiles of noKO/iKO at schizont stage (mean of triplicates)/late ring stage (mean of duplicates) around transcriptional landmark sites and corresponding GC-content profiles and selected zoom-ins. **f**, MNase-seq occupancy profiles of iKO/noKO around +1 nucleosome (based on the transcription start sites as determined from nucleosome dynamics) at schizont stage (mean of triplicates)/late ring stage (mean of duplicates).

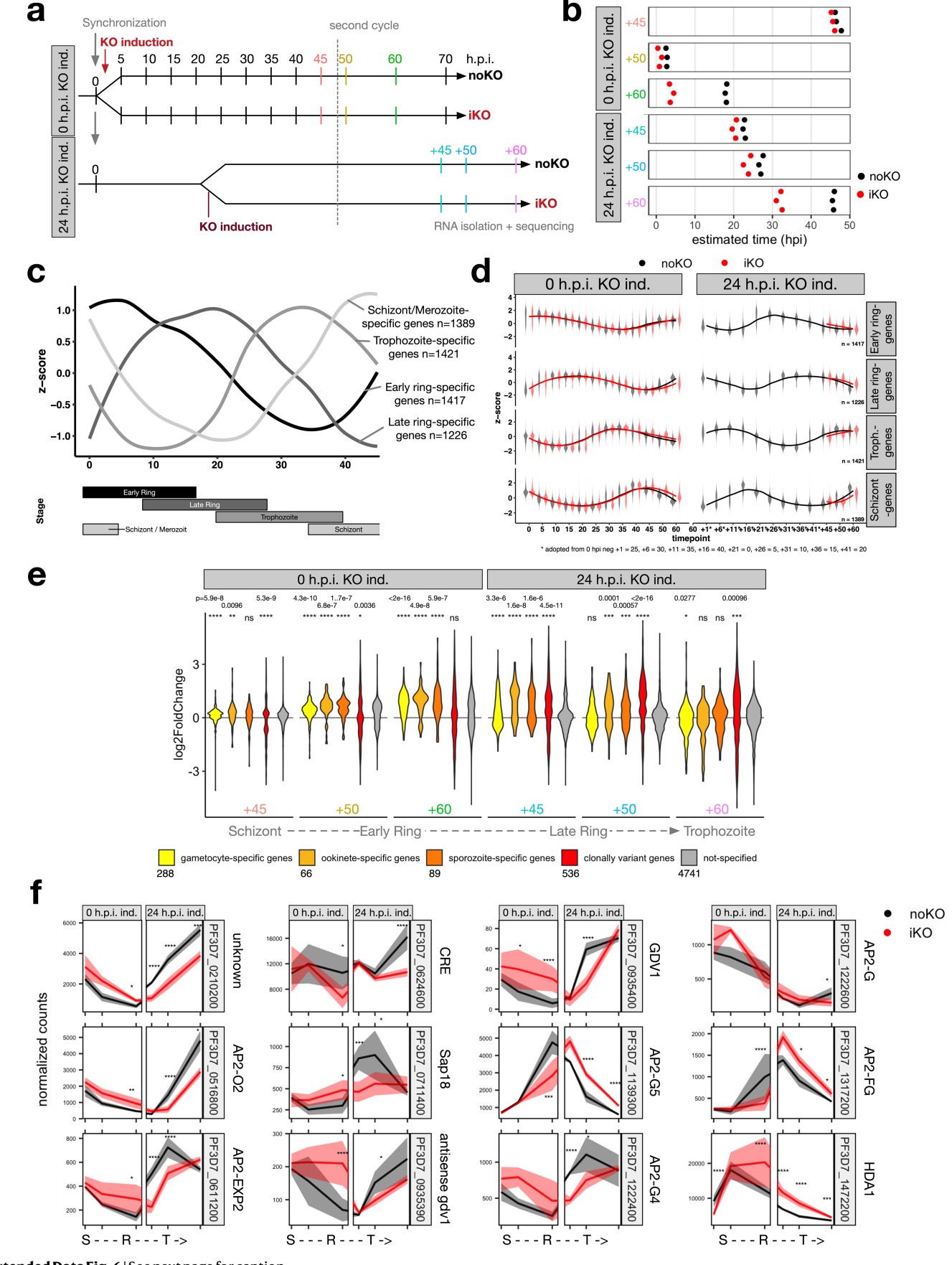

**Extended Data Fig. 6** | See next page for caption.

**Extended Data Fig. 6 | RNA-Seq analysis of *Pf*Snf2L-KO parasites revealed dysregulation of stage-specific genes.** *Pf*Snf2L-HA-KO parasites non-induced (noKO) and induced (iKO) at 0 and 24 h.p.i. were analysed +45/+50/+60 h later. All data shown are based on n = 3 biological replicates. **a**, Experimental setup of *Pf*Snf2L-KO timing and sampling for transcriptome analysis **b**, Cycle progression estimation of noKO/iKO RNA-seq samples of individual replicates shown in a. **c**, Z-score normalized gene expression over the life cycle of control parasites was grouped into four clusters. The average expression trajectory of all clusters follows a stage-specific pattern of up- and down regulation (top). The expression peak of each cluster is mainly associated with a single stage of the asexual life cycle (bottom). The clusters are named according to the associated stage: Numbers of Early ring- (n = 1417 genes), Late ring- (n = 1226 genes), Trophozoite- (n = 1421 genes) and Schizont/Merozoite-specific genes (n = 1226 genes) are indicated. **d**, Gene expression trajectories of the four clusters from b are compared between noKO (black) and iKO (red). The genes in the iKO samples are delayed in their up- or down-regulation compared to noKO after *Pf*Snf2L protein disappearance at timepoints +45 h, +50 h and +60 h. Z-score normalized genes in a cluster are shown for each time point and a Loess smoothed solid curve traces the expression over time for easier comparison of progression. To visualize the expected curve in the 24 h.p.i. experiment, the corresponding time points were taken from the 0 h.p.i. noKO samples. Accordingly, the depicted 24 h.p.i. +1 time point is the same as the 0 h.p.i. noKO 25 h.p.i. time point. The other time points are plotted as follows: +6 = 30, +11 = 35, +16 = 40, +21 = 0, +26 = 5, +31 = 10, +36 = 15, +41 = 20. **e**, Expression differences – log2(iKO/noKO) – at individual timepoints for stage-specific gene groups (Supplementary Table 2). **f**, Gene expression profiles (DESeq2 normalized) over individual timepoints of exemplary epigenetic regulators involved in gametocyte commitment. Mean is shown as line, SD is shown as transparent band. Significance is shown (two-tailed Student's t-test, unpaired, ns p > 0.05, * p < 0.05, ** p < 0.01, *** p < 0.005, ****<0.001). P-values for the six consecutive timepoints are: PF3D7_0210200: 0.206, 0.0128, 0.0204, 0.00682, 0.00573, 0.00482; PF3D7_0516800: 0.122, 0.0465, 0.124, 0.109, 0.0261, 0.0197; PF3D7_0611200: 0.426, 0.262, 0.156, 0.00334, 0.0276, 0.0157, PF3D7_0624600: 0.792, 0.922, 0.106, 0.88, 0.25, 0.0313; PF3D7_0711400: 0.666, 0.111, 0.106, 0.0206, 0.168, 0.241; PF3D7_0935390: 0.415, 0.331, 0.0491, 0.115, 0.0969, 0.238; PF3D7_0935400: 0.234, 0.182, 0.0639, 0.615, 0.00376, 0.102; PF3D7_1139300: 0.21, 0.745, 0.0424, 0.0226, 0.00891, 0.00614; PF3D7_1222400: 0.196, 0.0253, 0.332, 0.000978, 0.107, 0.818; PF3D7_1222600: 0.294, 0.0331, 0.569, 0.331, 0.167, 0.0888; PF3D7_1317200: 0.436, 0.388, 0.181, 0.00966, 0.0149, 0.0687; PF3D7_1472200: 0.0452, 0.697, 0.186, 0.0201, 0.0147, 0.0282.

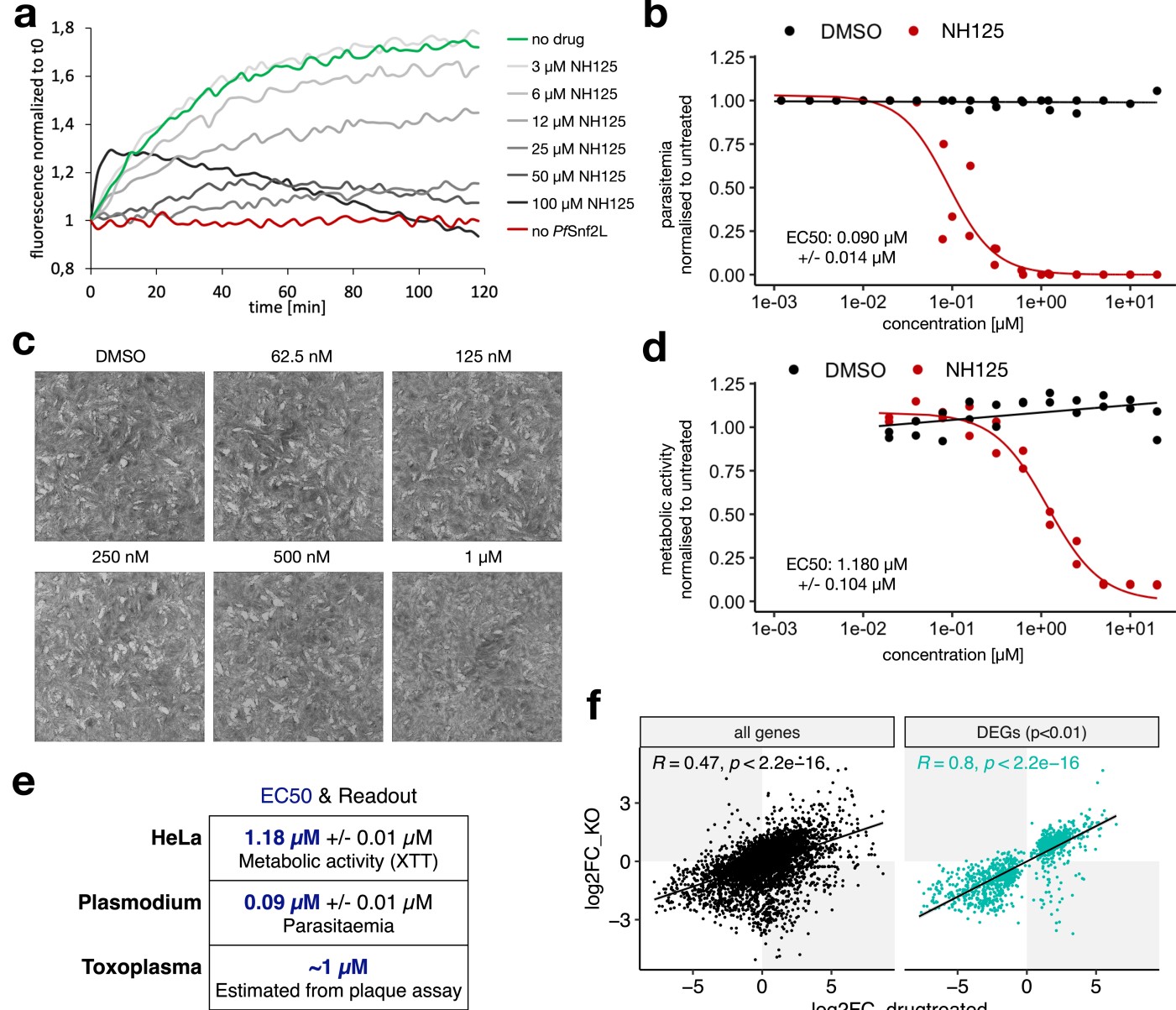

**Extended Data Fig. 7 | The *Pf*Snf2L inhibitor specifically kills *Plasmodium falciparum* and changes gene expression similar to *Pf*Snf2L-KO. a**, ATP hydrolysis kinetic of *Pf*Snf2L in presence of NH125 in varying concentrations (grey) with normalized readout of fluorescent ParM. A single dose response analysis with varying inhibitor concentrations is shown. *Pf*Snf2L (+DMSO) in absence of drug (green) serves positive control, samples without ATPase (red) as negative control. **b**, *Pf* Toxicity assay with parasitemia after 72 h in presence of varying concentrations of NH125 or DMSO. The results of n = 3 independent experiments is plotted and normalized to the parasitemia of untreated cells. The EC50 was determined by fitting dose response curve. **c**, Toxoplasma toxicity. Exemplary plaque assay of Toxoplasma gondii infected human foreskin fibroblasts incubated for 1 week before fixation and staining **d**, HeLa toxicity.

Metabolic activity of HeLa cells (monitored by XTT assay) in presence of varying concentration of NH125 or DMSO. The results of n = 2 independent experiments were plotted and normalized to untreated cells. EC50 determination as performed as in b. **e**, Summary of NH125 toxicity assay results, performed with HeLa, Plasmodium and Toxoplasma cells. The EC50 values and the corresponding readout methods are given. **f**, Correlation between gene expression changes of late *Pf*Snf2L-KO (+60 h.p.i.) and NH125 treatment cells (+35 h). All genes (left panel) and genes showing a significant change over both timecourse experiments (p < 0.01) (right panel) were analysed (significance is given according to a two-sided Pearson-test). DEG, Differentially Expressed Genes.

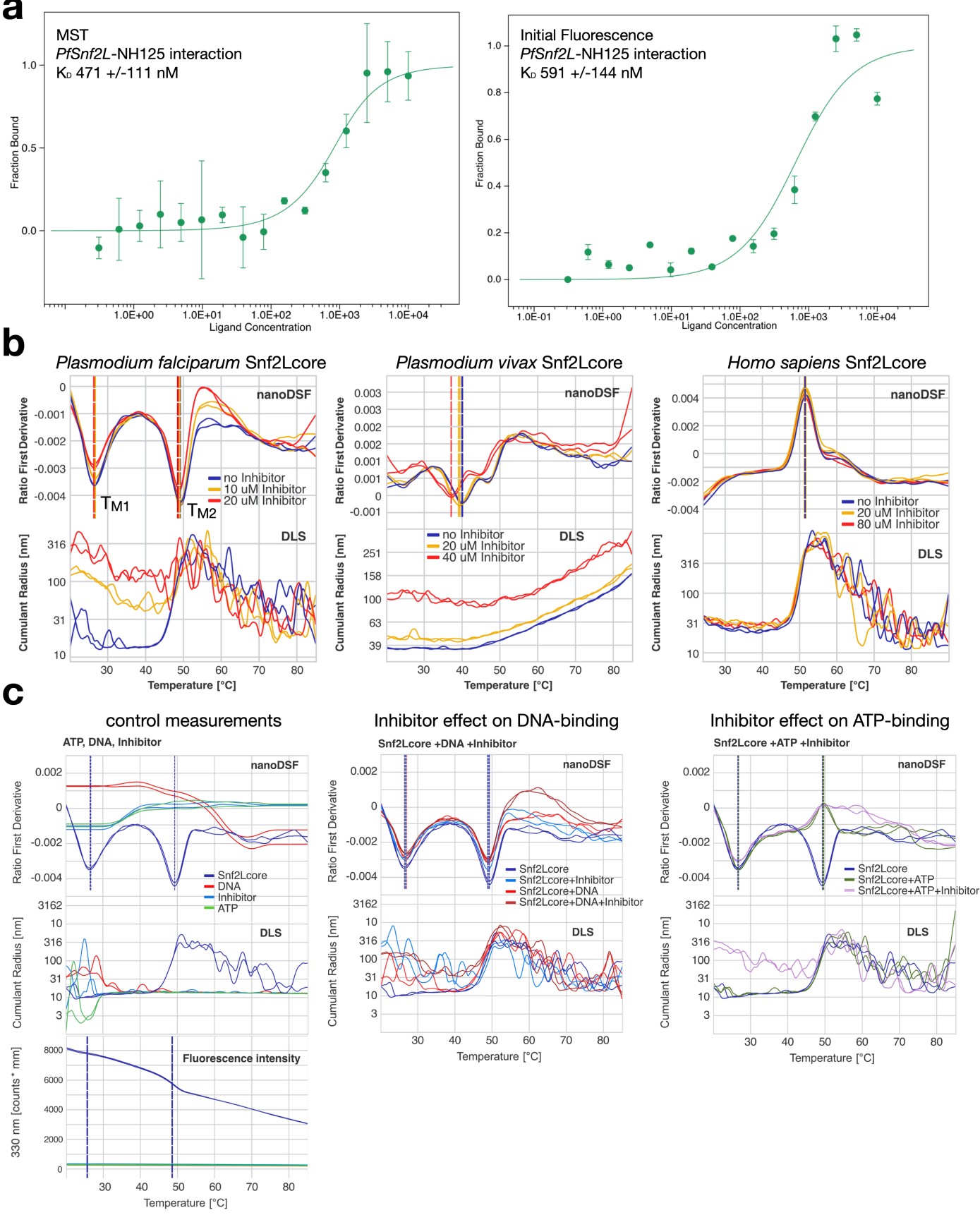

**Extended Data Fig. 8** | See next page for caption.

**Extended Data Fig. 8 | Biophysical characterization of NH125 shows *Pf*Snf2L-specific repression of ATP hydrolysis activity without affecting substrate binding. a**, Binding of NH125 to *Pf*Snf2L measured by Microscale Thermophoresis and Initial Fluorescence, using *Pf*Snf2L core enzyme (50 nM, labelled with a fluorescent His-Tag binder) and increasing concentrations of NH125. Mean ± s.d. of n = 3 independent replicates are shown. Left panel shows fluorescence changes upon thermophoresis and the estimated binding affinity. Right panel shows changes in initial fluorescence upon incubation with increasing concentrations of NH125. Binding curves are fitted and estimated $K_D$ values are shown. **b**, Differential scanning fluorimetry (nanoDSF, upper panel) and Dynamic Light Scattering (DLS, lower panel) analysis of the purified proteins *Pf*Snf2Lcore (10 μM, left panel), *Pv*Snf2Lcore (10 μM, mid panel), and *Hs*Snf2Lcore (10 μM, right panel), in the absence, or presence of the indicated concentrations of NH125. The curves show n = 2 biological replicates (same colour), as indicated. The upper panel shows the ratio of the first derivative of the nanoDSF fluorescence values measured at 330 and 350 nm, presenting the melting temperature of the individual proteins (marked as vertical dashed lines).

The lower panel shows the particle size of the same reaction, as measured by DLS. Particle sizes are given as cumulant radius over the increasing temperature. **c**, Thermal unfolding measurement of *Pf*Snf2Lcore (10 μM) in the presence of NH125 (20 μM) and DNA (20 μM) or ATP (1 mM). The left panel shows control measurements of the individual molecules *Pf*Snf2Lcore (blue), DNA (red), NH125 (light blue), and ATP (green). Quantification of NH125, DNA and ATP values shows nanoDSF and DLS curves without specific patterns (left panel). However, these values can be ignored, as the absolute fluorescence values are close to zero and present only background fluorescence (Fluorescence intensity, lowest panel on the left). Only *Pf*Snf2L shows significant fluorescence values (8000 fluorescence units at 20 °C). This means when performing measurements in the presence of *Pf*Snf2L, the NH125, DNA, and ATP values will not be detected, but only the fluorescence signal of the about 80-fold more intense *Pf*Snf2L will be recorded. The middle and right panels show the interaction of *Pf*Snf2L and NH125 with DNA, or ATP, as indicated. The combined signal changes in nanoDSF and DLS indicated the formation of trimeric complexes of *Pf*Snf2L + NH125 and ATP/DNA.

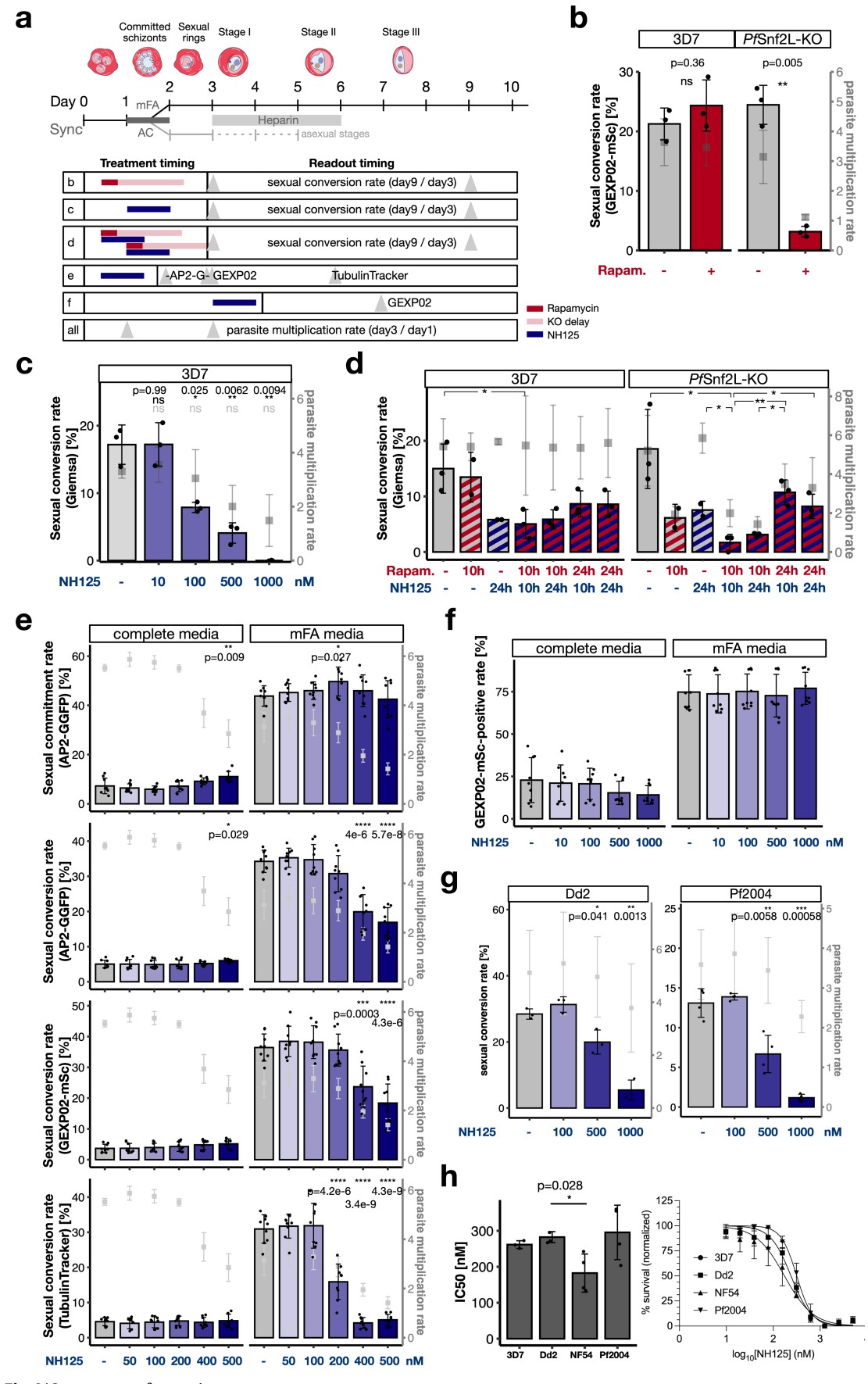

**Extended Data Fig. 9** | See next page for caption.

**Extended Data Fig. 9 | NH125 mediated *Pf*Snf2L inhibition and *Pf*Snf2L-KO block gametocyte development. a**, Schematic representation of gametocyte induction using minimal fatty acid (mFA) media 24–48 h.p.i. or Albumax + choline (AC) as control. Treatment timing and readout timing/method of the subsequent subfigures b-f are indicated. Plasmodia illustrations from NIAID NIH BIOART. **b**, Sexual conversion rate upon Rapamycin-treatment 24 h before gametocyte induction of 3D7 and *Pf*Snf2L-KO parasites, analogous to Fig. 3c (single data points and mean ± s.d. of triplicates are shown, two-tailed Student's t-test, paired, ns p > 0.05, * p < 0.05, ** p < 0.01). Parasite multiplication rates (PMR) on the right axis are shown in grey (mean ± s.d.). **c**, Sexual conversion rate upon treatment at 24 h.p.i. with different concentrations of NH125 (single data points and mean ± s.d. of triplicates are shown, two-tailed Student's t-test, paired, ns p > 0.05, * p < 0.05, ** p < 0.01). Parasite multiplication rates (PMR) on the right axis are shown as crosses (mean ± s.d.). **d**, Sexual conversion rate upon Rapamycin- and/or 100 nM NH125 treatment at 10/24 h.p.i. of 3D7 and *Pf*Snf2L-KO parasites, analogous to Fig. 4e (single data points and mean ± s.d. of triplicates are shown, two-tailed Student's t-test, paired, ns p > 0.05, * p < 0.05, ** p < 0.01, exact p-values see Fig. 4). Parasite multiplication rates (PMR) on the right axis are shown in grey (mean ± s.d.). **e**, Flow cytometry readout of sexual commitment rate via AP2-GGFP from schizonts at day 2 (panel 1), sexual conversion rate via AP2-GGFP from rings (panel 2), via GEXP02-mScarlet from sexual rings at day 3 (panel 3) and sexual conversion rate via TubulinTracker™

from gametocytes at day 6 (panel 4). Rates are calculated over day 2 parasitemia of the dual reporter NF54 parasites upon NH125 treatment with/without gametocyte induction, analogous to Fig. 4f (single data points and mean ± s.d. of triplicates are shown). Parasite multiplication rates (PMR) on the right axis are shown in grey (mean ± s.d.). **f**, Flow cytometry readout of GEXP02-mScarlet-positive rate of dual reporter NF54 parasites upon treatment (2 days post gametocyte induction) with different concentrations of NH125 (single data points and mean ± s.d. of technical triplicates per biological triplicates are shown, two-tailed Student's t-test, paired, ns p > 0.05). **g**, Flow cytometry readout of sexual conversion rates (day 5 gametocytemia over day 2 parasitemia) post gametocyte induction (mFA media 24/48 h.p.i.) for Dd2 and Pf2004 parasites exposed to different NH125 concentrations (single data points and mean ± s.d. of triplicates are shown). Gametocytemia was measured by TubulinTracker™ (Dd2) or by the gametocyte-specific expression of the tdTomato reporter in transgenic Pf2004 parasites (164-tdTomPf2004[2],). Parasite multiplication rates (PMR) are shown on the right axis in grey (mean ± s.d.). **h**, IC50 values for NH125 (72 h treatment) on four different parasite strains (left panel, single data points and mean ± s.d. of n = 3/n = 4 biological replicates are shown). Normalized dose response curves are shown in the right panel (mean ± s.d.). NF54 describes the dual reporter NF54 parasite line and Pf2004 describes the single reporter Pf2004 parasite line.

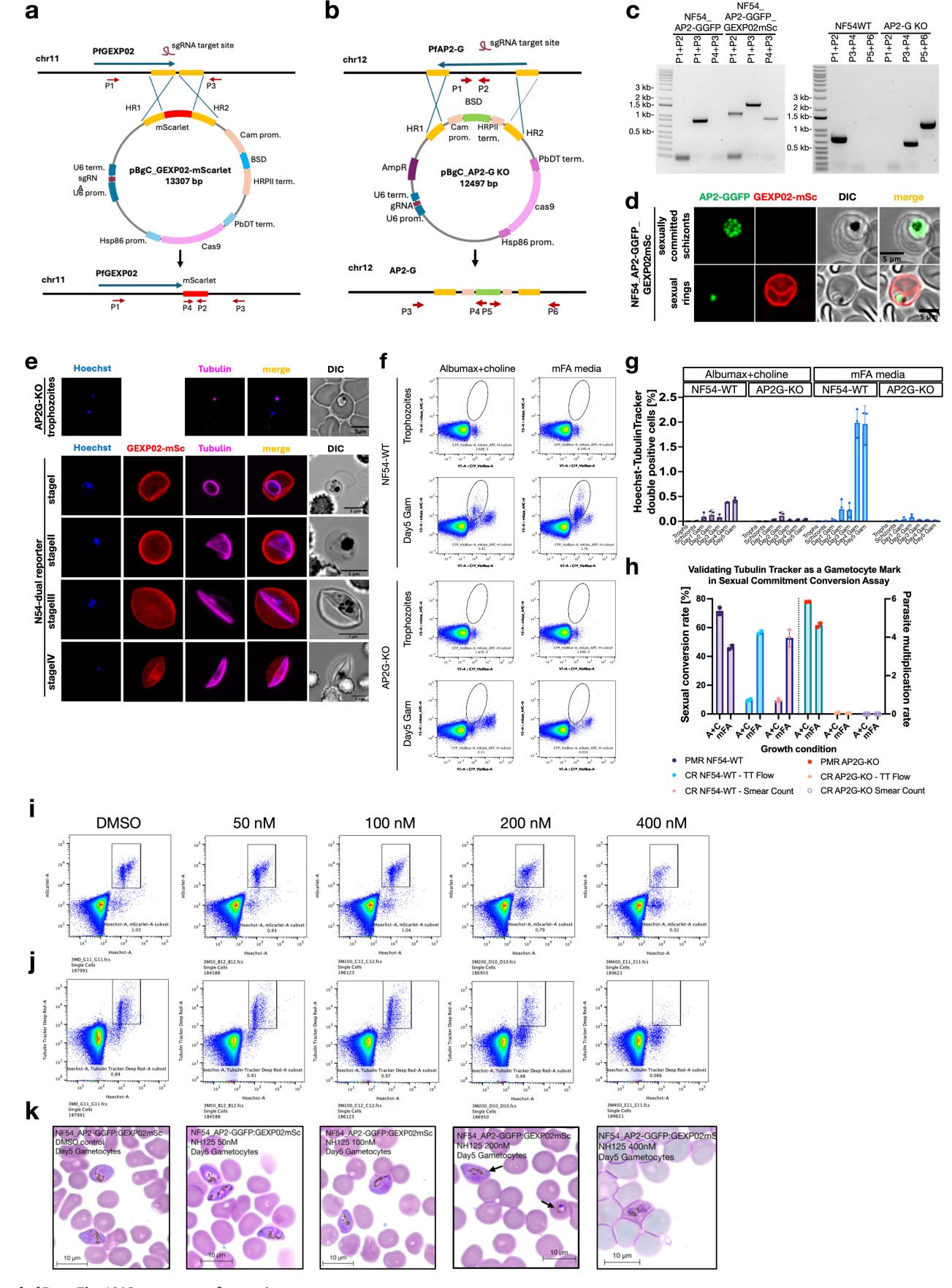

**Extended Data Fig. 10** | See next page for caption.

**Extended Data Fig. 10 | Establishment of a dual reporter parasite lines to study sexual commitment, conversion and gametocyte development.**
**a**, Schematic representation for generation of dual reporter NF54 parasites. Wildtype locus, episomal plasmid and locus after endogenous integration as well as primers P1-4 used for genotyping PCR are depicted. Abbreviations. GEXP02: Gametocyte Exported Protein 02, HR: homologous region, sgRNA: single guide RNA, BSD: Blasticidin S deaminase gene. **b**, Schematic representation for generation of AP2-G KO parasites. Wildtype locus, plasmid and locus after endogenous integration as well as primers P1-6 used for genotyping PCR are depicted. Abbreviations. HR: homologous region, sgRNA: single guide RNA, BSD: Blasticidin S deaminase gene. **c**, Confirmation of homologous recombination by PCR genotyping with primers indicated in a/b. n = 1 replicate was performed **d**, Immunofluorescence analysis of dual reporter NF54 parasites at sexually committed schizont (50 h.p.i.) and ring stage (16 h.p.i.) using AP2-G GFP and GEXP02-mScarlet reporters. Scale bar: 5 µm. One representative image of n = 3 independent replicates is shown. **e**, Immunofluorescence analysis of NF54 dual reporter and AP2G-KO parasites at different stages with Hoechst, GEXP02-mScarlet and TubulinTracker™ Deep Red as marker. Scale bar: 5 µm. One representative image of n = 3 independent replicates is shown. **f**, Exemplary scatter plots from flow cytometry analysis of NF54 WT and AP2G-KO parasites at trophozoite/Day5 gametocyte stage upon gametocyte induction (mFA media treatment versus complete media). TubulinTracker™ Deep Red is used as gametocyte reporter. **g**, Quantification of Hoechst-TubulinTracker™ double- positive cells by flow cytometry in different stages across gametocyte development (parasites and treatments as in f). Mean +/− SEM of biological triplicates are shown. **h**, Sexual Conversion Rate (CR) calculated using smear counts and TubulinTracker™ Deep Red, readout on Day5 Gametocytes in NF54 WT vs AP2G-KO parasites upon gametocyte induction (mFA media treatment versus complete media). Mean +/− SEM of biological triplicates are shown. **i**, Scatter plots from flow cytometry analysis of dual reporter NF54 parasites upon NH125 treatment. GEXP02-mScarlet as an endogenous reporter (and Hoechst-staining) is used to detect Day5 gametocytes with an intact and identifiable tubulin cytoskeleton (corresponds to Day 6 in Extended Data Fig. 9a schematic). **j**, Scatter plots as in i, with TubulinTracker™ Deep Red used to detect Day5 gametocytes. **k**, Representative Giemsa stain images of Day5 Gametocytes as in i,/j,. Scale bar: 10 µm. Arrows at 200 nM NH125 indicate one intact Day5 gametocyte (i.e., TubulinTracker™ Deep Red [TT] positive) and one arrested gametocyte (i.e., TT negative). One representative field of view of n = 1 replicate is shown.

# Reporting Summary

## Statistics

For all statistical analyses, confirm that the following items are present in the figure legend, table legend, main text, or Methods section.

| n/a | Confirmed | |
|---|---|---|
| ☐ | ☒ | The exact sample size (*n*) for each experimental group/condition, given as a discrete number and unit of measurement |
| ☐ | ☒ | A statement on whether measurements were taken from distinct samples or whether the same sample was measured repeatedly |
| ☐ | ☒ | The statistical test(s) used AND whether they are one- or two-sided<br>*Only common tests should be described solely by name; describe more complex techniques in the Methods section.* |
| ☒ | ☐ | A description of all covariates tested |
| ☒ | ☐ | A description of any assumptions or corrections, such as tests of normality and adjustment for multiple comparisons |
| ☐ | ☒ | A full description of the statistical parameters including central tendency (e.g. means) or other basic estimates (e.g. regression coefficient) AND variation (e.g. standard deviation) or associated estimates of uncertainty (e.g. confidence intervals) |
| ☐ | ☒ | For null hypothesis testing, the test statistic (e.g. *F*, *t*, *r*) with confidence intervals, effect sizes, degrees of freedom and *P* value noted<br>*Give P values as exact values whenever suitable.* |
| ☒ | ☐ | For Bayesian analysis, information on the choice of priors and Markov chain Monte Carlo settings |
| ☒ | ☐ | For hierarchical and complex designs, identification of the appropriate level for tests and full reporting of outcomes |
| ☐ | ☒ | Estimates of effect sizes (e.g. Cohen's *d*, Pearson's *r*), indicating how they were calculated |

*Our web collection on statistics for biologists contains articles on many of the points above.*

## Software and code

Policy information about availability of computer code

| Data collection | not applicable |
|---|---|
| Data analysis | All software used is cited in the Material and Methods section and the Supplementary Information |

For manuscripts utilizing custom algorithms or software that are central to the research but not yet described in published literature, software must be made available to editors and reviewers. We strongly encourage code deposition in a community repository (e.g. GitHub). See the Nature Portfolio guidelines for submitting code & software for further information.

## Data

Policy information about availability of data

All manuscripts must include a data availability statement. This statement should provide the following information, where applicable:
- Accession codes, unique identifiers, or web links for publicly available datasets
- A description of any restrictions on data availability
- For clinical datasets or third party data, please ensure that the statement adheres to our policy

RNA- and MNase-Sequencing data generated for this study have been deposited in the Gene Expression Omnibus (GEO) database (accession number GSE228949). Sequencing analysis pipeline is available at https://github.com/SimHolz/Watzlowik_et_al_2023. Proteomic data have been deposited to the ProteomeXchange (PXD041155).

# Research involving human participants, their data, or biological material

Policy information about studies with human participants or human data. See also policy information about sex, gender (identity/presentation), and sexual orientation and race, ethnicity and racism.

| | |
|---|---|
| Reporting on sex and gender | not applicable |
| Reporting on race, ethnicity, or other socially relevant groupings | not applicable |
| Population characteristics | not applicable |
| Recruitment | not applicable |
| Ethics oversight | not applicable |

Note that full information on the approval of the study protocol must also be provided in the manuscript.

# Field-specific reporting

Please select the one below that is the best fit for your research. If you are not sure, read the appropriate sections before making your selection.

☒ Life sciences          ☐ Behavioural & social sciences          ☐ Ecological, evolutionary & environmental sciences

For a reference copy of the document with all sections, see nature.com/documents/nr-reporting-summary-flat.pdf

# Life sciences study design

All studies must disclose on these points even when the disclosure is negative.

| | |
|---|---|
| Sample size | not applicable |
| Data exclusions | no data was excluded |
| Replication | In vivo experiments were performed as triplicates and in vitro experiments were performed at least three times |
| Randomization | not applicable |
| Blinding | not applicable |

# Reporting for specific materials, systems and methods

We require information from authors about some types of materials, experimental systems and methods used in many studies. Here, indicate whether each material, system or method listed is relevant to your study. If you are not sure if a list item applies to your research, read the appropriate section before selecting a response.

## Materials & experimental systems

| n/a | Involved in the study |
|---|---|
| ☐ | ☒ Antibodies |
| ☐ | ☒ Eukaryotic cell lines |
| ☒ | ☐ Palaeontology and archaeology |
| ☐ | ☒ Animals and other organisms |
| ☒ | ☐ Clinical data |
| ☒ | ☐ Dual use research of concern |
| ☒ | ☐ Plants |

## Methods

| n/a | Involved in the study |
|---|---|
| ☐ | ☒ ChIP-seq |
| ☐ | ☒ Flow cytometry |
| ☒ | ☐ MRI-based neuroimaging |

## Antibodies

| | |
|---|---|
| Antibodies used | rat anti-HA Roche<br>mouse anti-Enolase Prof GK Jarori, Tata Institute for Fundamental Research<br>rabbit anti-Gap45 Jones et al. 2006<br>rabbit anti-Ama1 Collins et al. 2009 |

| Validation | as described in the cited publications |

# Eukaryotic cell lines

Policy information about cell lines and Sex and Gender in Research

| Cell line source(s) | Plasmodium falciparum 3D7 with DiCre, Blackman Laboratory<br>Toxoplasma gondii RHdeltaHX, Markus Meissner Laboratory<br>Human Foreskin fibroblasts, ATCC, SCRC-1041<br>HeLa cells, ATCC, CCL-2 |
| Authentication | not applicable |
| Mycoplasma contamination | tested, no contamination |
| Commonly misidentified lines<br>(See ICLAC register) | not applicable |

# Animals and other research organisms

Policy information about studies involving animals; ARRIVE guidelines recommended for reporting animal research, and Sex and Gender in Research

| Laboratory animals | not applicable |
| Wild animals | not applicable |
| Reporting on sex | not applicable |
| Field-collected samples | not applicable |
| Ethics oversight | not applicable |

Note that full information on the approval of the study protocol must also be provided in the manuscript.

# Plants

| Seed stocks | not applicable |
| Novel plant genotypes | not applicable |
| Authentication | not applicable |

# ChIP-seq

## Data deposition

☒ Confirm that both raw and final processed data have been deposited in a public database such as GEO.

☐ Confirm that you have deposited or provided access to graph files (e.g. BED files) for the called peaks.

| Data access links<br>*May remain private before publication.* | accession number GSE228949 |
| Files in database submission | not applicable |
| Genome browser session<br>(e.g. UCSC) | not applicable |

## Methodology

| Replicates | 2 biological replicates |
| Sequencing depth | at least 20fold coverage |
| Antibodies | HA antibody |
| Peak calling parameters | no peak calling used |

| Data quality | Read quality was assesed by FastQC, filtered for multimapped reads; MapQ=1 |
| --- | --- |
| Software | Trimmomatic, FastQC, BWA-MEM, DeepTools, BamCompare |

# Flow Cytometry

## Plots

Confirm that:

☒ The axis labels state the marker and fluorochrome used (e.g. CD4-FITC).

☒ The axis scales are clearly visible. Include numbers along axes only for bottom left plot of group (a 'group' is an analysis of identical markers).

☒ All plots are contour plots with outliers or pseudocolor plots.

☒ A numerical value for number of cells or percentage (with statistics) is provided.

## Methodology

| Sample preparation | *Describe the sample preparation, detailing the biological source of the cells and any tissue processing steps used.* |
| --- | --- |
| Instrument | *Identify the instrument used for data collection, specifying make and model number.* |
| Software | *Describe the software used to collect and analyze the flow cytometry data. For custom code that has been deposited into a community repository, provide accession details.* |
| Cell population abundance | *Describe the abundance of the relevant cell populations within post-sort fractions, providing details on the purity of the samples and how it was determined.* |
| Gating strategy | *Describe the gating strategy used for all relevant experiments, specifying the preliminary FSC/SSC gates of the starting cell population, indicating where boundaries between "positive" and "negative" staining cell populations are defined.* |

☒ Tick this box to confirm that a figure exemplifying the gating strategy is provided in the Supplementary Information.

