## [Peer Review File · Nature]

Plasmodium blood stage development requires the chromatin remodeler Snf2L

Corresponding Author: Professor Gernot Längst

Version 0:

Reviewer comments:

Referee #1

(Remarks to the Author)

Summary of the key results:

The authors show that PfSNF2L functions as an ATP-dependent chromatin-remodeller that move and place nucleosomes in vitro and found in complexes that regulate transcription and chromatin structure.

Inducible knockouts of this essential gene, begin to manifest observable phenotypes upon subsequent merozoite egress & re-invasion and full developmental arrest in the next asexual development cycle.

The developmental phenotypes in the subsequent cycle are associated with an increase in global chromatin accessibility / nucleotide positioning and changes in gene expression of many genes.

Lastly, they used recombinant SNF2L to identify tool compounds that inhibit *P. falciparum* growth in the low nanomolar range with moderate (10x) selectivity.

Treatment with the inhibitor phenocopied the KO in terms of morphology and gene expression. Both KO and the inhibitor also suppressed sexual commitment.

This study was well designed, carefully executed, and is presented clearly in its language and figures.

I have no concerns regarding robustness, validity or reliability.

While the authors do their best to highlight novel features of PfSNF2L, the manuscript shows that, given the evolutionary distance to model organisms, its main structural features and function are well conserved.

As a chromatin remodeller that modulates the access of transcription factors to their binding sites PfSNF2L broadly carries out the same tasks it homologs do in other eukaryotes.

While SNF2L is essential for parasite growth and progression of transcriptional program that underlies asexual replication in blood stages, the manuscript claim that PfSNL2F "governs" just-in-time transcription is poorly supported in so far as "governs" implies control/regulation.

"Allows" or "Facilitates" might be better choices. Just-in-time transcription has to be driven has to be driven by sequence-specific regulators that bind to cognate motifs/marks and then either recruit/repress RNA polymerases or provide/block access to other factors that perform this function.

Little suggests that PfSNF2L acts in a sequence specific manner but rather it is part of many complexes the contain transcription factors or histone readers that will target it to specific loci.

Its role in facilitating sexual commitment is of interest since SNF2L is one of the earliest differentially expressed genes in AP2-G positive cells.

Since most gene expression is affected upon loss of SNF2L function, differential gene expression of GDV1 doesn't make SNF2L an upstream regulator of commitment any more than RNA pol II.

Just because something is important or even essential for a process does not inherently make it a regulator of the process. Without additional data, such claims should be removed from line 182 and elsewhere.

Suggested improvements:

Unfortunately, SNF2L's importance for sexual commitment and gametocyte development was only explored in very limited

fashion. Is it required just during the commitment cycle? In gametocytes?

The cKO experiments also indicate that SNF2L is not required for the first 7 days of gametocyte maturation since the readout was morphology on Day 8.

Though since the induction of the knockout is imperfect it would be important to check that the proportion of SNF2L KO is similar in the remaining gametocytes than on Day 2, otherwise the residual GCs might come from the parasites where SNF2L was not deleted.

The number of independent replicates must be indicated for each panel. Inter-experimental errors should be provided for all numerical data.

1: replace "governs" with a term that doesn't imply a fundamental regulatory role. (see discussion above)

40: Remove "DNA". No convincing evidence for functional DNA modifications exists in *P. falciparum*. The suggested DNA methyltransferase is an ortholog of the confusingly named DNMT2 family and has been conclusively demonstrated the methylate RNA like other DNMT2s. *Pf* has no orthologs of true DNA methyltransferases.

63/64 and beyond: remove "non-conserved" and similar statements. If can't be a homolog and "non-conserved". At best it is "divergent" though Fig 1 and subsequent functional analyses show that nearly all of the key functional domains are conserved, as is its function.

99: "reveals a role in nucleosome assembly and chromatin dependent regulation of gene expression" Fig1b-d shows a role in nucleosome assembly but their presence in complexes with known transcriptional regulators only "implies" or "suggests" a role in the regulation of gene expression by "guilt through association".

109: Even KO induction at 24h clearly impacts the re-invasion rate since the parasitemia is substantially reduced relative to no induction in Fig 2A.

132: are the "substantial" change significant?

160: "In total, 75% of genes are DEG..." I found this sentence confusing. If 200 DEGs in schizonts is 75% of expressed genes does that mean that only 267 genes are expressed in schizonts? That seems quite low.

177: "crucial" means critical/key/essential but gametocytes still form. Unless you can demonstrate that all the iKO failed in the residual GCs, this should be replaced with a weaker adjective.

244: "mimics" was confusing here since the inhibitor has an immediate effect while the effect in the iKO is seen in the next cycle. This makes sense since the iKO requires depletion of existing protein while the inhibitor does not. It might be worth stating this outright.

Fig 2b/d/e: are these changes significant?

Fig 2e: x-axis label should be "relative proportion of events" not "genome"

Fig 3d: For ease of the reader please re-define "NFR" in the legend.

Fig 4b: The Toxo IC50 adds little since *Toxoplasma* resides in fibroblast with an active nucleus and the IC50 against mammalian cells is the same as that against Toxo, making it difficult to separate effects on the parasite from those on the host.

Björn F.C. Kafsack

Referee #2

(Remarks to the Author)

In this compelling study, Watzlowik and colleagues seek to understand the role of PfSnf2L at the biochemical, genomic, and phenotypic level, culminating in the identification of a PfSnf2L specific inhibitor. PfSnf2L has 30% homology to Human Snf2L with particular conservation in its ATPase core, but lacks conservation in the autoregulatory and substrate binding regions. The authors show that this enzyme binds and assembles nucleosomes and exhibits nucleosome stimulated ATPase activity and remodeling activity. In vitro biochemical studies suggest that PfSnf2L has increased ATPase hydrolysis activity and reduced remodeling activity to Human Snf2L. Immunoprecipitation mass spectrometry studies reveal interactions with a number of nuclear chromatin regulators, but the authors do not explore these interacting proteins in mechanistic detail. Inducible knockout models identify a role for PfSnf2L in inducible gene regulation during *Plasmodium falciparum* life cycle, particularly affecting development at the late ring stage. The authors examine nucleosome spacing by MNase-seq to identify changes in occupancy, expanded (fuzziness) and shifted peak calls, with generally more open chromatin in PfSnf2L late ring knockouts. Gene expression analysis revealed hundreds of DEGs affecting multiple developmental and physiological pathways and showing perturbed induction of timing specific genes. PfSnf2L function at the nucleosome free region (+1 nucleosome) in promoters was highly correlated with gene expression changes particularly at highly expressed genes. Finally, due to the lack of conservation between PfSn2L and HsSnf2L, the authors undertook a chemical inhibitor screen to identify a specific chemical inhibitor which inhibits PfSnf2L ATPase activity and remodeling activity with 10 fold specificity for *Plasmodium* versus HeLa toxicity or *Toxoplasma* infection. The identified NH125 compound has a similar effect on *Plasmodium* replication and gametocyte conversion rate as the inducible knockout, as well as transcription timing and gene expression fold change, suggesting on target activity.

This is a very nice study that seeks to functionally define a new ISWI remodeler with limited conservation to other Snf2L members, including lack of complex subunits. The profound effect of the knockout on *Plasmodium* replication and timing-specific transcription make it an attractive drug target. Indeed, the authors take this next step in identifying a preclinical inhibitor with notable specificity. This is a complete and compelling study, employing multiple assays to definitely demonstrate the mechanistic and functional roles of this newly defined remodeler.

I don't have any concerns with this well-conducted and interesting study. One point of interest—the authors in several places suggest that the PfSnf2L remodeler may have evolved to handle the demands of the AT-rich Plasmodium genome. It would be of interest to conduct ATPase activity, assembly, and remodeling assays with AT-rich sequences to determine if this enzyme is superior at binding and remodeling at these sequences.

Referee #3

(Remarks to the Author)

The chromatin based 'epigenetic' gene regulation in Plasmodium falciparum is of particular interest from an evolutionary point of view, but also because the identification of parasite-specific regulation may pave the way for therapeutic intervention with malaria.

Watzlowik and colleagues present an impressive study on the nucleosome remodeling ATPase SNF2L, distantly related to mammalian SNF2 enzymes. Such enzymes are known for their general nucleosome spacing functions, but their ability to slide nucleosomes also allows DNA-binding regulators to access their sequence recognition elements. The enzyme was identified through sequence similarity searches and apparently has not been studied to date. The current paper contains a most comprehensive analysis ranging from biochemical assays of nucleosome remodeling, chromatin interactions and nucleosome mapping in vivo, loss-of-function phenotypes during development and, finally presents an inhibitor that inactivates PfSNF2L at a 10-fold lower dose than the human remodeler SNF2L. I have been impressed by the breadth and depth of the study and particularly liked the transcriptome analysis suggesting a shift in gene expression timing upon iKO.

However, I also spotted a few weaknesses that might be addressed to improve clarity and impact of the paper. I will restrict myself to comments on aspects related to chromatin, which is my area of expertise.

Major issues

1. Nomenclature. The enzyme of study (PF3D7_1104200) is called PfSNF2L following an earlier suggestion of Ji and Arnot (ref 25). However, in extended data (ED) 1a this enzyme groups with dmISWI in a branch distinct from other SNF2L enzymes. There, in fact, the enzyme PF3D7_0216000 is the closest relative to mammalian SNF2 enzymes. The authors could rationalize again their nomenclature. The paper by Ji and Arnot was published in 1997, when the full scope of remodelers was perhaps not clear.

Later in the work (ED 7e), the authors mention 'ISWI' (PF3D7_0624600) as one of the genes that is regulated by 1104200. This enzyme separates with a second protein at the bottom of the evolutionary tree. Why is this called ISWI and how does it relate to the other enzymes?

It is also unclear which enzyme is called ISWI in ED1b? Clearly, the evolutionary analysis clearly requires more sophistication.

On a sideline: how is the enzyme under study related to the gametocyte SNF2 in Kaneko et al. (doi: 10.1073/pnas.2303432120)? This paper should be cited and discussed in context.

2. 'ISWI' (PF3D7_0624600) is one of the putative enzymes that is drastically down-regulated upon KO of 1104200. ISWI interacts with SNF2L as seen by IP-MS (Figure 1f). If 0624600 is a remodeling enzyme as well, it is possible, that the consequences of SNF2L KD are mediated to some extent (largely?) through ISWI. Can that be ruled out?

3. The characterization of SNF2L as a nucleosome remodeling ATPase in vitro essentially rests on its activity on vertebrate nucleosomes. The authors use nucleosomes from chicken cows and human, depending on experiments (ED2c, etc).

In Figure 1b the source of histones is not mentioned.

Given the sequence similarities of histones this is no major concern, but it shows that different assays were done at different times by different people.

However, it is of concern that most of the reactions are NOT performed with plasmodium histones, despite the fact that the authors have histone octamers available (ED1c,d). The question that arises, is whether plasmodium nucleosomes can be remodeled by SNF2L?

The only experiment where Pf nucleosomes are used is in Figure 1d, but the quality of this experiment is poor, relative to the others. It is unclear how nucleosomes are defined (in contrast to the 'undefined histone complexes' marked by triangles). How would hsSNF2L behave in this assay?

4. It is mentioned that nucleosomes are reconstituted on 'a nucleosome positioning sequence', but no-where indicated what that sequence is. I assume it is akin to the 'Widom 601' sequence, but this should be spelled out. Do Pf histones position on such a sequence? The authors imply a lot of similarity between PfSNFL and better-known enzyme, but do not show assays with plasmodium proteins to convince that such extrapolation is appropriate.

5. The authors argue that a major mechanism through which SNF2L activates genes is changes in nucleosome positioning. The nucleosome positions shown in Figure 2D differ between schizont and late ring stage (by the criteria used to show the effect of iKO). Is that due to the expression state of the underlying genes?

ED6d suggest that the main positioning clue in the late ring stage is the transition between general AT-rich DNA and GC-rich coding sequences (AUG and stop codon). Interestingly, the positions of the nucleosome at the boundary are hardly affected by KO. The induced knock-out mainly leads to reduced occupancy of nucleosomes outside of reading frames. Does SNF2L affect nucleosome assembly at AT-rich DNA rather than positioning?

6. I did not find the description of the ATPase kinetic experiment (ED 8a) described in the methods. Does NH125 affect hsSNF2L in such a purified assay?

Minor issues

7. Figure 1c: To which extent are the ATPase activities stimulated by free DNA? A conclusion about 'nucleosome-stimulated' ATPase can only be done relative to free DNA. This is important since nucleosome density varies with CG-content in the plasmodium genome and the enzyme requires linker DNA (ED2e) to bind to nucleosomes.

8. Figure 3D: I do not understand how the NFR width is determined (and could not find the description in the methods).

Version 1:

Reviewer comments:

Referee #1

(Remarks to the Author)

The revisions further improved an already strong manuscript.

Unfortunately, the my main question/concern remains:

Is PfSNF2L, a specific regulator of gene expression that controls or "governs" the activation and silencing of specific *P. falciparum* expression programs in a sequence specific manner

OR is it an essential component of the general machinery required for the silencing and activation of most genes?

As mentioned in the rebuttal, the authors have evidence for specific regulation by some chromatin remodelers in higher eukaryotes but evidence for such a remodeling code in malaria parasites, which diverged 850 million years ago, is limited.

In fact, three points raised in the manuscript speak against the idea of SNF2L and other chromatin remodelers having a specific regulatory roles in *P. falciparum*.

First, the fact malaria parasites only encode a single orthologs from each family of ATPase remodelers (Fig. S1a) and that many components of remodeling complexes are absent (line 50) suggests that only core functions are conserved without the functional diversification of complexes that enables specialized regulation of specific genes sets.

The second argument for a general function rather than a specific one, is the large number of differentially expressed genes observed in the iKO.

In the rebuttal, the author state that they observed differential gene expression for 75% of genome's 5600.

This indicates that all or nearly all transcription is affected thereby strongly indicating a general role in transcription, not a specific one, and speaks against a primarily sequence-dependent specific role.

The third argument against a specific role in just in-time transcription, is that the transcriptional program underlying asexual replication is able to advance even in iKO parasite, though somewhat delayed (Fig 3a., Fig.S7c).

Finally, the rate of sexual commitment is clearly altered in iKO parasites. However, since sexual commitment requires the activation of two otherwise silent regulators (*gdv1* and *ap2-g*) and loss of PfSNFL broadly alters gene activation and repression generally (Line 164).

Based on these observations, SNF2L appears to be a general regulator of gene expression rather than specifically "governing" key steps.

Every cog in clock may be essential to its function but that doesn't mean that each cog "governs"/regulates what time it shows. Otherwise, being a "regulator" becomes meaningless.

Björn Kafsack

Referee #2

(Remarks to the Author)

The authors have addressed my comments and I congratulate them on a beautiful study.

Referee #3

(Remarks to the Author)

The authors extensive revised their manuscript in response to the comments of editors and reviewer. They clarified the

confusion revolving around the nomenclature and characterized the biochemical remodeling reactions with Ph histones and on AT-rich DNA. It is fairly obvious, that classical remodeling assays yield only limited information with these atypical nucleosomes. Given the documented expertise of the authors (also highlighted in previous work), I assume that the results are 'as good as it gets'.

I appreciate the wealth of new data on the developmental effects of KO and inhibitor treatment and on the binding of the inhibitor to the enzyme. While the biochemistry is consistent with the in vivo effects on nucleosome positioning and differential gene expression, it is these in vivo data (including inhibitor effects) that make the manuscript a nice publication that is suitable for 'Nature'.

I have no concerns about publication of the manuscript.

Version 2:

Reviewer comments:

Referee #1

(Remarks to the Author)

I appreciate the author's further clarification and apologize for misreading the sexual commitment / conversion results. As the authors correctly state, my arguments on that point were indeed invalid.

The authors state in their rebuttal that "show that PfSnf2L is a stage-specific regulator of gene expression" but this is not quite correct. They show that SNF2L is necessary for stage-specific expression, which inherently requires the activation or silencing of these genes.

They clearly show nucleosome repositioning by PfSNF2L is required for just-in-time transcription, i.e. to activate non-expressed genes and to silence already expressed genes. Continued expression of constitutively expressed genes involves no nucleosome repositioning and therefore does not require PfSNF2L.

This is similar to functions of complexes containing its homolog in yeast, where it was shown to be required to reposition nucleosomes to activate genes required for mating or genes required for growth on sucrose. Similarly, complexes containing the fruit fly and mammalian homologs have been shown to reposition nucleosomes to allow activation or silencing of genes.

But the article should not say that it GOVERNS or CONTROLS this process as something else decides which promoters SNF2L is recruited to. It is required for stage-specific expression but it is not what determines the stage-specificity of individual genes.

SNF2L not what directs the activation or silencing of these genes. Something else generally recruits SNF2L containing complexes to promoters and hence it cannot be said to GOVERN this process. Instead the title and text could say that SNF2L is NECESSARY/REQUIRED FOR or MEDIATES just-in-time transcription.

Björn Kafsack

Watzlowik et al. answers to the reviewer comments

The answers to the reviewers are given as blue text.

We are grateful for the reviewers' comments and suggestions on our manuscript. We have addressed all the issues that were raised by performing additional experiments. The response letter explains the results.

Editor

SNF2L's importance for sexual commitment and gametocyte development (referee 1; this will be the important issue in our view), to provide additional data to support the role of SNF2L in "governing" just-in-time gene expression and gametocytogenesis rather than showing it as one of the many regulators essential for gametocytogenesis (referee 1), to check the remodeling activity towards AT-rich DNA (referees 2 and 3), and to use *Plasmodium falciparum* histones for in vitro characterization (referee 3). Given the reservation of referee 1 regarding novelty, anything to strengthen the SNF2L inhibitor part would be appreciated (for instance, does it outperform the existing drugs or what is its advantage over existing drugs?).

As requested by the Editor, we fully addressed all concerns raised by the reviewers by performing and including additional experiments. Furthermore, as requested by the Editor, we performed additional biophysical experiments to characterize the mechanistic interaction of the inhibitor with remodeling enzymes. These experiments are described at the end of our reply to the reviewers.

Referee #1 (Remarks to the Author):

Summary of the key results:

The authors show that PfSNF2L functions as an ATP-dependent chromatin-remodeller that move and place nucleosomes in vitro and found in complexes that regulate transcription and chromatin structure.

Inducible knockouts of this essential gene, begin to manifest observable phenotypes upon subsequent merozoite egress & re-invasion and full developmental arrest in the next asexual development cycle.

The developmental phenotypes in the subsequent cycle are associated with an increase in global chromatin accessibility / nucleotide positioning and changes in gene expression of many genes.

Lastly, they used recombinant SNF2L to identify tool compounds that inhibit *P. falciparum* growth in the low nanomolar range with moderate (10x) selectivity.

Treatment with the inhibitor phenocopied the KO in terms of morphology and gene expression. Both KO and the inhibitor also suppressed sexual commitment.

This study was well designed, carefully executed, and is presented clearly in its language and figures.

I have no concerns regarding robustness, validity or reliability.

We appreciate the reviewer's favourable assessment of our research.

While the authors do their best to highlight novel features of PfSNF2L, the manuscript shows that, given the evolutionary distance to model organisms, its main structural features and function are well conserved.

As a chromatin remodeller that modulates the access of transcription factors to their binding sites PfSNF2L broadly carries out the same tasks it homologs do in other eukaryotes.

While SNF2L is essential for parasite growth and progression of transcriptional program that underlies asexual replication in blood stages, the manuscript claim that PfSNL2F "governs" just-in-time transcription is poorly supported in so far as "governs" implies control/regulation.

"Allows" or "Facilitates" might be better choices. Just-in-time transcription has to be driven has to be driven by sequence-specific regulators that bind to cognate motifs/marks and then either recruit/repress RNA polymerases or provide/block access to other factors that perform this function.

Little suggests that PfSNF2L acts in a sequence specific manner but rather it is part of many complexes the contain transcription factors or histone readers that will target it to specific loci.

We (especially Gernot Längst) disagree with the view of the reviewer on the function of chromatin remodeling complexes (CRC). There is sufficient data showing that CRCs have much more regulatory function than RNA Polymerase II. We previously showed and suggested that CRCs shape the nucleosomal landscape with respect to the underlying sequence, thereby modulating the accessibility of DNA and regulating gene expression. Regulation of transcription requires the cooperation of sequence-specific binding factors and their specific recruitment to the chromatinized templates. CRCs are not simple facilitators of DNA access but also recognise the sequence/structure features of DNA/Chromatin and do move nucleosomes to remodeler-specific sites. The CRCs act as gene- and context-specific activators or repressors that sometimes allow binding of sequence-specific DNA factors and, in other contexts, repress their DNA binding by positioning the nucleosome over the binding site. We published this concept in Rippe et al. (PNAS 2007) and subsequent manuscripts. The cell type-specific combination of hundreds of different CRC complexes shapes a specific chromatin landscape that allows gene-specific factor binding and cell type-specific gene expression. We call this concept the remodeling code.

Also *PfSnf2L* has its specific remodeling code that positions nucleosomes with respect to the sequence context. The sequence-specific positioning is evident from the iKO experiments and the MNase-seq maps, which show altered nucleosome positions. By altering the nucleosome positions, genes are mistakingly activated or repressed upon loss of *PfSnf2L*. We suggest that the many different remodeling enzymes work as an overarching regulatory mechanism, establishing a nucleosome positioning landscape that allows selective access to DNA binding factors. These enzymes are essential machines in the regulatory network and exhibit sequence-specific remodeling features.

The *in vitro* remodeling experiments on the artificial NPS sequences are not ideal for recapitulating the sequence-specific remodeling features of *PfSnf2L*, but in Extended Data Figure 3 different nucleosome positioning patterns between *PfSnf2L*, *HsSnf2L*, and *Chd4* can be seen. The experiments presented in this manuscript were not designed to reveal sequence specific nucleosome positioning, but it can already be observed in the presented

examples. Together with the additional experiments provided in this revision, we suggest to keep the term govern.

Its role in facilitating sexual commitment is of interest since SNF2L is one of the earliest differentially expressed genes in AP2-G positive cells.

Since most gene expression is affected upon loss of SNF2L function, differential gene expression of GDV1 doesn't make SNF2L an upstream regulator of commitment any more than RNA pol II.

Just because something is important or even essential for a process does not inherently make it a regulator of the process. Without additional data, such claims should be removed from line 182 and elsewhere.

Suggested improvements:

Unfortunately, SNF2L's importance for sexual commitment and gametocyte development was only explored in very limited fashion. Is it required just during the commitment cycle? In gametorings?

The cKO experiments also indicate that SNF2L is not required for the first 7 days of gametocyte maturation since the readout was morphology on Day 8.

Though since the induction of the knockout is imperfect it would be important to check that the proportion of SNF2L KO is similar in the remaining gametocytes than on Day 2, otherwise the residual GCs might form the parasites where SNF2L was not deleted.

Thank you. This is an extremely important point, and we agree with the reviewer that additional data were required to define the role of *PfSnf2L* in gametocytogenesis. To do so, we teamed up with two renowned experts in the field of malaria gametocytogenesis and differentiation, Manuel Llinás (The Pennsylvania State University, USA) and Matthias Marti (University of Zurich, Switzerland). To address the request of the reviewer, we now include additional experiments, as shown in the new Figures 4e and f, Extended Data Figures 11a, b, c, d, e and f and Extended Data Figures 13a, b and c. The new data address the role of *PfSnf2L* from both sides, by investigating gametocytogenesis under knockout conditions and inhibition through drug treatment. In addition to using the standard method of counting for *PfSnf2L*-KO parasites, we made use of a new, highly sensitive dual reporter assay allowing for differentiation between commitment and early gametocyte differentiation. Generation and validation of the dual reporter NF54 parasites is described in the Supplementary Information and Extended Data Figure 12 a, b, c, d, e, f. Combining this transgenic parasite line with two endogenously tagged markers and additional tubulin staining for independent quantification of gametocyte numbers, we now demonstrate that early gametocyte development, but not commitment is affected by drug treatment (Fig. 4f). Using various setups with carefully coordinated treatment conditions and readout timings, we show that *PfSnf2L* exerts a crucial function downstream of AP2-G activity (Extended Data Fig. 11 a, e, f). The effect of impaired gametocyte development is specific (i.e. not due to a general toxic effect), since PMR remains unaffected under the experimental conditions used.

Importantly, all involved labs reproduced the gametocyte development phenotype, and we now demonstrate that the inducible KO is hypersensitive to the inhibitor NH125 (Extended Data Figure 11d, Fig 4e), since the conversion rate is significantly reduced in comparison to drug treatment and KO only. Combining both treatments, we can ensure that *PfSnf2L* is efficiently depleted in the system, resulting in a dramatic reduction in the proportion of gametocytes when compared to single treatments.

We updated the manuscript text at lines 271-286 (new additions in the manuscript text are given in blue color) and included the additional section “Sexual commitment conversion assay – detailed method section and additional data interpretation” in the Supplementary Information describing our conclusions in detail.

The number of independent replicates must be indicated for each panel. Inter-experimental errors should be provided for all numerical data.

Previously, we commented on the number of replicates in the Materials and Method sections. Now we have moved this information into the text and figure legends, as suggested by the reviewer for Figures 2a, 2d, 3a, 3c, 3d, 4c, ED4b, ED6a, ED6b, ED6c, ED6d, ED6e, ED6h, ED7b, ED7d, ED7e, ED7f, ED8, ED8b, ED8c, ED9a. Interexperimental errors were added to Figure ED6d and to all new numerical data.

1: replace "governs" with a term that doesn't imply a fundamental regulatory role. (see discussion above)

In light of the newly added data and a discussed above, we suggest keeping the term “govern”.

40: Remove "DNA/". No convincing evidence for functional DNA modifications exists in *P. falciparum*. The suggested DNA methyltransferase is an ortholog of the confusingly named DNMT2 family and has been conclusively demonstrated the methylate RNA like other DNMT2s. *Pf* has no orthologs of true DNA methyltransferases.

The sentence in the manuscript, as well as the cited literature, refers to eukaryotes in general. It is now described more clearly as “eukaryotic epigenetic mechanisms”. (line 46)

63/64 and beyond: remove "non-conserved" and similar statements. If can't be a homolog and "non-conserved". At best it is "divergent" though Fig 1 and subsequent functional analyses show that nearly all of the key functional domains are conserved, as is its function.

PfSnf2L is now described as divergent enzyme instead of using the term “non-conserved”. (line 70)

99: "reveals a role in nucleosome assembly and chromatin dependent regulation of gene expression" Fig1b-d shows a role in nucleosome assembly but their presence in complexes with known transcriptional regulators only "implies" or "suggests" a role in the regulation of gene expression by "guilt through association".

The statement was changed by adding “imply” as recommended. (109)

109: Even KO induction at 24h clearly impacts the re-invasion rate since the parasitemia is substantially reduced relative to no induction in Fig 2A.

Parasitemia upon *PfSnf2L*-knockout induced at 24 hpi is indeed slightly reduced in the following early ring stage in Fig. 2a. However, this effect is not significant when compared to the non-induced control. To specifically address this observation, we performed an invasion assay with the knockout induced at 0 and 24 hours. As shown in Extended Data Figure 6c, the number of newly infected red blood cells is only significantly reduced for early *PfSnf2L*-KO, but not for KO induced 24 hpi. Combining the findings of those two assays with the observed delay of ~40-45 hours until the KO induction leads to the disappearance of *PfSnf2L* on protein level (Extended Data Figure 5d), we speculate that at the time the assay was performed sufficient *PfSnf2L* levels were still present in the majority of parasites, which were consequently still able to invade normally. Only in a few parasites *PfSnf2L* reached a „critical“ low level, leading to an invasion defect. This result is also in good agreement with the observed hypersensitivity of induced iKO-parasites to NH125 (see above).

132: are the "substantial" change significant?

In short, the answer is yes. The observed differences are highly significant. Below, we describe the nucleosome selection and significance determination of the data presented in the manuscript.

The tool used to detect changes in nucleosome position shifts, fuzziness and occupancy (DANPOS) uses a chi-squared test to assess changes in fuzziness and a Poisson test to assess changes in occupancy. We further implemented an additional script that simulates random nucleosome positions to adjust the resulting p-values to a false discovery rate (FDR) of nucleosome shifts. In our manuscript, we analyse only nucleosomes detected with high confidence, by selecting the 20% best-positioned (=lowest fuzziness score) nucleosomes of the combined stages with respect to the noKO condition. Using the FDR values provided with a significance threshold of $p < 0.05$, we identified 27376/29354 out of 38497 (71.1%/76.3%) high confidence nucleosomes and 5267/7330 out of 15364 (34.3%/47.7%) high confidence nucleosomes with significant occupancy/fuzziness changes for late ring stage and schizont stage respectively. We then used an unbiased approach to define thresholds for an even more stringent selection. Thresholds were selected by using a slope cutoff of > 1 , applied to a Loess regression fit to the normalized variance of the specific feature (occupancy, fuzziness, position shift) against the normalized variance rank of each nucleosome position. This resulted in cutoffs of $p < 9.787 \cdot 10^{-29}$, $p < 3.212 \cdot 10^{-30}$ for the respective occupancy and fuzziness FDR values and a cutoff of at least a 40 bp shift for nucleosomes with position shifts (Extended Data Figure 7d). As this is not a common procedure that applies standard statistical thresholds, we decided not to write "significant" and opted for the term "substantial" changes.

To provide a better impression of the variation and significance of the experiments, we changed the presentation of Figure 2d and Extended Data Figure 7b. We now include not only the pooled mean of the MNase-Seq map, but show the minimum/maximum interval of all replicates. Furthermore, the increased changes of the nucleosome map in late ring stage compared to schizont stage, further strengthens our statement that we are observing biological and not experimental variations. In addition, we now include a new Extended Data

Figure 7c detailing the quantitative changes of nucleosome fuzziness, occupancy and shifts in the two stages.

To clarify the MNase-seq data analysis and interpretation we included the additional section “MNase-Sequencing – detailed data analysis and interpretation” to the Supplementary Information.

160: "In total, 75% of genes are DEG..." I found this sentence confusing. If 200 DEGs in schizonts is 75% of expressed genes does that mean that only 267 genes are expressed in schizonts? That seems quite low.

The statement is now changed in the manuscript to make the point clear (line 169f.):
Analysing all differentially expressed genes in every developmental stage across the full erythrocytic cycle and adding them up, results in 75% of all 5600 genes being misregulated significantly in at least one time point. In contrast, stage-specific - at schizont stage - only 200 of all genes are misregulated, while at trophozoite stage we identified about ~2000 DEG. Across all sexual stages, about ~4200 genes are expressed differentially.

177: "crucial" means critical/key/essential but gametocytes still form. Unless you can demonstrate that all the iKO failed in the residual GCs, this should be replaced with a weaker adjective.

Even though the inducible KO system is very efficient, it is nearly impossible to achieve 100 % protein knockout. In addition, due to the delicate timing of gametocytogenesis gene regulation, a slight delay in *PfSnf2L* gene excision may cause a small proportion of functional *PfSnf2L* protein, still being active at the critical timepoint of gametocyte-specific gene regulation. This likely scenario would explain the small proportion of residual gametocytes despite *PfSnf2L*-KO. To address the concerns about the leftover enzyme, we combined KO induction with *PfSnf2L* inhibition. This additional data clearly shows a synergistic effect of both disruption methods, again emphasising the impact of treatment/induction timing. In light of the newly added data, we suggest keeping the term crucial. The new experiments are shown in Figure 4e and Extended Data Figures 11d, and the new findings are described in the manuscript in lines 272-286.

244: "mimics" was confusing here since the inhibitor has an immediate effect while the effect in the iKO is seen in the next cycle. This makes sense since the iKO requires depletion of existing protein while the inhibitor does not. It might be worth stating this outright.

The inhibitor has an immediate effect on the activity of *PfSnf2L*, whereas the knockout results in delayed effects due to the slow kinetics of protein degradation. Nonetheless, the immediate effects of the inhibitor perfectly mirror the knockout effects when taking into account the time delay. This is most evident in Figure 4d, detecting a very high overlap between inhibitor- and knockout-mediated gene expression changes ($R=0.8$), showing that the same genes are affected in the respective developmental stages. This high overlap made us use the term mimic. We agree that this is not the best term in this context, as the inhibitor has no effect on its own. Instead, we changed the term to “reproduces” (line 258).

Fig 2b/d/e: are these changes significant?

Yes, the changes are significant. A detailed description is given above (see 132:).

Fig 2e: x-axis label should be "relative proportion of events" not "genome"

The mislabeling of the x-axis was changed, as suggested.

Fig 3d: For ease of the reader please re-define "NFR" in the legend.

The abbreviation NFR is now explained in the legend.

Fig 4b: The Toxo IC50 adds little since *Toxoplasma* resides in fibroblast with an active nucleus and the IC50 against mammalian cells is the same as that against Toxo, making it difficult to separate effects on the parasite from those on the host.

The Figure shows the specificity of the drug, saying that it has no effect on the related species *Toxoplasma*. Indeed, this results in the same IC50, as seen for mammalian cells. The informative value of this assay needs to be interpreted with caution. *Toxoplasma* and *Plasmodium* toxicity assays are only partially comparable, since the host cells – fibroblast and erythrocytes – are potentially different in their ability to take up NH125. Comparing the toxicity in the intracellular context just indicates that NH125 is specific for *Plasmodium falciparum*. In agreement with the reviewers comment, the figure is now moved to the Extended Data Figures (Extended Data Figure 9c, e). A set of novel biophysical experiments, now document the species specificity of NH125, suggesting very low affinity binding for *Plasmodium vivax* Snf2L and no binding towards human Snf2L (Extended Data Figure 10).

Björn F.C. Kafsack

Referee #2 (Remarks to the Author):

In this compelling study, Watzlowik and colleagues seek to understand the role of PfSnf2L at the biochemical, genomic, and phenotypic level, culminating in the identification of a PfSnf2L specific inhibitor. PfSnf2L has 30% homology to Human Snf2L with particular conservation in its ATPase core, but lacks conservation in the autoregulatory and substrate binding regions. The authors show that this enzyme binds and assembles nucleosomes and exhibits nucleosome stimulated ATPase activity and remodeling activity. In vitro biochemical studies suggest that PfSnf2L has increased ATPase hydrolysis activity and reduced remodeling activity to Human Snf2L. Immunoprecipitation mass spectrometry studies reveal interactions with a number of nuclear chromatin regulators, but the authors do not explore these interacting proteins in mechanistic detail. Inducible knockout models identify a role for PfSnf2L in inducible gene regulation during *Plasmodium falciparum* life cycle, particularly affecting development at the late ring stage. The authors examine nucleosome spacing by MNase-seq to identify changes in occupancy, expanded (fuzziness) and shifted peak calls, with generally more open chromatin in PfSnf2L late ring knockouts. Gene expression analysis revealed hundreds of DEGs affecting multiple developmental and physiological pathways

and showing perturbed induction of timing specific genes. PfSnf2L function at the nucleosome free region (+1 nucleosome) in promoters was highly correlated with gene expression changes particularly at highly expressed genes. Finally, due to the lack of conservation between PfSn2L and HsSnf2L, the authors undertook a chemical inhibitor screen to identify a specific chemical inhibitor which inhibits PfSnf2L ATPase activity and remodeling activity with 10 fold specificity for Plasmodium versus HeLa toxicity or Toxoplasma infection. The identified NH125 compound has a similar effect on Plasmodium replication and gametocyte conversion rate as the inducible knockout, as well as transcription timing and gene expression fold change, suggesting on target activity.

This is a very nice study that seeks to functionally define a new ISWI remodeler with limited conservation to other Snf2L members, including lack of complex subunits. The profound effect of the knockout on Plasmodium replication and timing-specific transcription make it an attractive drug target. Indeed, the authors take this next step in identifying a preclinical inhibitor with notable specificity. This is a complete and compelling study, employing multiple assays to definitely demonstrate the mechanistic and functional roles of this newly defined remodeler.

I don't have any concerns with this well-conducted and interesting study. One point of interest—the authors in several places suggest that the PfSnf2L remodeler may have evolved to handle the demands of the AT-rich Plasmodium genome. It would be of interest to conduct ATPase activity, assembly, and remodeling assays with AT-rich sequences to determine if this enzyme is superior at binding and remodeling at these sequences.

We appreciate the reviewer's favourable evaluation of our manuscript and have carried out the recommended experiments.

PfSnf2L acts on chromatin reconstituted on a highly AT-rich genome in *Plasmodium falciparum*. The question arises as to whether *PfSnf2L* has evolved to act on this specific substrate.

First, we quantified the binding affinity of *PfSnf2L* to short (20 bp) DNA fragments by EMSA and MST. We decided to use such short DNA with low binding affinity to avoid avidity effects by the multiple DNA binding sites of remodeling enzymes, thereby scrutinising sequence specificity. As shown in EMSA (new Extended Data Figure 2e), *PfSnf2L* does bind preferentially to the AT-rich DNA over the GC-rich DNA. The experiments were performed as competitive binding assays with differentially labelled DNA molecules to allow the mixing of the substrates in the test tube. The EMSA results were confirmed by Microscale Thermophoresis (MST) experiments using fluorescently labelled protein with the non-labelled DNA molecules. We observe binding of *PfSnf2L* to AT-rich DNA (20 bp) with an affinity of about 7 μ M, whereas no binding to the GC-rich DNA could be observed. Indeed, our experiments suggest an evolutionary optimisation of *PfSnf2L* towards an AT-rich genome. These experiments are now included in the manuscript (lines 69-74, marked in blue). The experiments are described in detail in the novel Supplementary Information "In vitro characterization of *PfSnf2L* – additional data interpretation" and included as new Extended Data Figures 2e and f.

Second, we included additional *in vitro* remodeling experiments, using different nucleosomal templates harbouring a strong nucleosome positioning sequence (NPS). We use the NPS sequence to allow site-specific nucleosome assembly not only with the human/bovine

histones, but also with the *Pf* histones. We previously showed that *Pf* histone octameres do not position well on DNA lacking the strong NPS sequence (Silberhorn et al., 2016). Using this strategy, we start with similar nucleosomal templates and evaluate the activity of *Pf* vs *Hs*Snf2L, as suggested by the reviewer. The novel experiments and the detailed description can be found in the Supplementary Information “In vitro characterization of *Pf*Snf2L – additional data interpretation “. In summary, the experiments show that *Hs*Snf2L has a higher remodelling activity (new Extended Data Figure 3a and b), and this is not dependent on the nucleosomal substrate. *Pf*Snf2L remodels recombinant human histone octamers and recombinant *Pf* octamers, reconstituted on the same position of a NPS containing template, with similar efficiency (new Extended Data Figure 3c). However, the remodeling outcome is different for the two types of histones octamers. Whereas *Pf*Snf2L moves the human octamers to discrete positions on the template, the *Pf* octamer positioning is highly diffuse. In summary, the human enzyme is more efficient and consumes less ATP per moved nucleosome. The novel experiments are mentioned in the manuscript (lines 69-74, marked in blue).

Third, we addressed nucleosome assembly and remodeling on AT-rich DNA templates. This was a tough task. The reviewer may be familiar with the fact that nucleosomes can hardly be formed on AT-rich DNA (Iyer Struhl 1995 Embo; Struhl 1985 PNAS; Yuan Rando 2005 Science; Segal 2008 PlosCompBiol). Furthermore, we previously showed that *Plasmodium falciparum* histone octamers prefer to assemble on GC-rich DNA, like the human octamers (Silberhorn et al., 2016). We selected 8 different genomic regions of *Pf* and prepared DNA templates with varying AT-content. From these sequences, we were able to reconstitute 5 into nucleosomes. Most of these nucleosomes were highly unstable and did not show a discrete positioning pattern. With two of these templates, we were able to perform nucleosome remodeling experiments with *Pf*Snf2L and the highly active human Chd4 remodeling enzyme for comparison (new Extended Data Figure 3d,e). Surprisingly, we see remodeling of the AT-rich templates with *Pf*Snf2L but not with Chd4. We do not observe remodeling reactions as beautiful, as on the classic remodeling templates, but convincing ATP-dependent nucleosome repositioning can be observed. As only two templates were compared, we do not conclude that human enzymes fail to remodel AT-rich templates, but we conclude that *Pf*Snf2L is capable of doing so. The experiments are now mentioned in the manuscript (lines 69-74, marked in blue), described in detail in the Supplementary Information “In vitro characterization of *Pf*Snf2L – additional data interpretation “, and the experiments are amended as Extended Data Figure 3d,e.

Referee #3 (Remarks to the Author):

The chromatin based ,epigenetic‘ gene regulation in *Plasmodium falciparum* is of particular interest from an evolutionary point of view, but also because the identification of parasite-specific regulation may pave the way for therapeutic intervention with malaria.

Watzlowik and colleagues present an impressive study on the nucleosome remodeling ATPase SNF2L, distantly related to mammalian SNF2 enzymes. Such enzymes are known for their general nucleosome spacing functions, but their ability to slide nucleosomes also allows DNA-binding regulators to access their sequence recognition elements. The enzyme was identified through sequence similarity searches and apparently has not been studied to date. The current paper contains a most comprehensive analysis ranging from biochemical

assays of nucleosome remodeling, chromatin interactions and nucleosome mapping in vivo, loss-of-function phenotypes during development and, finally presents an inhibitor that inactivates *PfSnf2L* at a 10-fold lower dose than the human remodeler SNF2L. I have been impressed by the breadth and depth of the study and particularly liked the transcriptome analysis suggesting a shift in gene expression timing upon iKO.

However, I also spotted a few weaknesses that might be addressed to improve clarity and impact of the paper. I will restrict myself to comments on aspects related to chromatin, which is my area of expertise.

Major issues

1. Nomenclature. The enzyme of study (PF3D7_1104200) is called *PfSNF2L* following an earlier suggestion of Ji and Arnot (ref 25). However, in extended data (ED) 1a this enzyme groups with dmISWI in a branch distinct from other SNF2L enzymes. There, in fact, the enzyme PF3D7_0216000 is the closest relative to mammalian SNF2 enzymes. The authors could rationalize again their nomenclature. The paper by Ji and Arnot was published in 1997, when the full scope of remodelers was perhaps not clear.

The reviewers comment regarding the nomenclature was addressed by including a detailed section “Phylogenetic analysis – additional data interpretation” in the Supplementary information with additional data interpretation of the phylogenetic analysis and by revising the branch labels in the Extended Data Figure 1a, b. Furthermore, the specific requests of the reviewer are answered as follows:

PF3D7_1104200 was historically called *PfSnf2L* based on the similarity to the human *Snf2L* enzyme, which belongs to the ISWI-family. In-depth phylogenetic analysis confirms the evolutionary proximity to the ISWI subgroup but no clear homology to one of the ISWI members (*Snf2L* or *Snf2H* (*Hs*) / *Isw1* or *Isw2* (*Sc*)). This is in accordance with the observation that PF3D7_1104200 is the only ISWI family enzyme in *Plasmodium falciparum* and potentially evolutionary separated before ISWI-subdivision in higher eukaryotes. The name of PF3D7_1104200, *PfSnf2L*, was not changed for consistency and comparability.

At this point it is important to clarify that the ISWI-subfamily enzymes, even though called *Snf2L* and *Snf2H*, are not related to the *Snf2*-subfamily. The *Snf2*-subfamily comprise enzymes with species-specific names like *Brm* (*Dm*), *Brg1* (*Hs*), *STH* (*Sc*), *gSNF2* (*Pb*), etc. (see Extended Data Table 3), and show the closest similarity with PF3D7_0216000.

Later in the work (ED 7e), the authors mention ‘ISWI’ (PF3D7_0624600) as one of the genes that is regulated by 1104200. This enzyme separates with a second protein at the bottom of the evolutionary tree. Why is this called ISWI and how does it relate to the other enzymes?

According to PlasmoDB, PF3D7_0624600 was called ISWI without a source publication or a conclusive basis for argumentation. According to Bryant et al. 2020 and our phylogenetic analysis, there is no evolutionary proximity to the ISWI group and any other subfamily. Due to no additional orthologs being identified beyond Apicomplexa, we propose an Apicomplexan evolutionary development for this enzyme. For comparability with existing literature, the historical name “ISWI” is not changed, but is avoided in the manuscript. For that reason, ISWI was also replaced for PF3D7_0624600 in Fig. 4 f.

It is also unclear which enzyme is called ISWI in ED1b? Clearly, the evolutionary analysis clearly requires more sophistication.

The labels in Extended Data Figure 1 a, b are revised and detailed information (species-specific names as well as subfamilies) is added to Extended Data Table 3. The unclear name of PF3D7_0624600 – ISWI – is avoided in the manuscript, as described above.

On a sideline: how is the enzyme under study related to the gametocyte SNF2 in Kaneko et al. (doi: 10.1073/pnas.2303432120)? This paper should be cited and discussed in context.

The study mentioned by the reviewer analyzes the CRE gSNF2, the *Plasmodium berghei* ortholog of PF3D7_0216000, which can be classified as a Snf2-subfamily member, a distinct class of remodeling enzymes. The findings of Kaneko et al. suggest a function of gSNF2 only in late gene expression, proposing a function in gametocyte differentiation, without any function in early stages. The study is now cited in the manuscript, supporting the important role of CREs for stage-dependent gene expression regulation.

2. 'ISWI' (PF3D7_0624600) is one of the putative enzymes that is drastically down-regulated upon KO of 1104200. ISWI interacts with SNF2L as seen by IP-MS (Figure 1f). If 0624600 is a remodeling enzyme as well, it is possible, that the consequences of SNF2L KD are mediated to some extent (largely?) through ISWI. Can that be ruled out? –

The two remodelers do interact and may cooperate in the regulatory network. However, we can clearly state that the observed effects are due to a function of *PfSnf2L*, as the effects of the *PfSnf2L* knockout are mirrored by the addition of the *PfSnf2L* inhibitor NH125. Supporting this, another study previously investigated the phenotype of "ISWI"-PF3D7_0624600-knockdown parasites (Bryant et al., 2020, ref. 34 in the manuscript). They found PF3D7_0624600 to be a transcriptional activator of *var* genes and didn't report any developmental defects or any impact on gametocytogenesis as we do for *PfSnf2L*-KO. We cannot rule out that KO of *PfSnf2L* has a downstream effect on PF3D7_0624600, but considering the deviating phenotypes, we suggest that *PfSnf2L* effects are not mediated by "ISWI".

3. The characterization of SNF2L as a nucleosome remodeling ATPase in vitro essentially rests on its activity on vertebrate nucleosomes. The authors use nucleosomes from chicken cows and human, depending on experiments (ED2c, etc).

In Figure 1b the source of histones is not mentioned.

The source of histones is now mentioned in the Figure legend.

Given the sequence similarities of histones this is no major concern, but it shows that different assays were done at different times by different people.

However, it is of concern that most of the reactions are NOT performed with plasmodium histones, despite the fact that the authors have histone octamers available (ED1c,d). The question that arises, is whether plasmodium nucleosomes can be remodeled by SNF2L?

The only experiment where Pf nucleosomes are used is in Figure 1d, but the quality of this experiment is poor, relative to the others. It is unclear how nucleosomes are defined (in contrast to the 'undefined histone complexes' marked by triangles. How would hsSNF2L behave in this assay?

We now included additional experiments addressing the activity of *Pf*Snf2L compared to *Hs*Snf2L, also using *Pf* histone octameres and AT-rich templates reconstituted with *Pf* octameres. For a detailed description, see the comments to reviewer 2.

4. It is mentioned that nucleosomes are reconstituted on 'a nucleosome positioning sequence', but no-where indicated what that sequence is. I assume it is akin to the 'Widom 601' sequence, but this should be spelled out. Do Pf histones position on such a sequence? The authors imply a lot of similarity between PfSNFL and better-known enzyme, but do not show assays with plasmodium proteins to convince that such extrapolation is appropriate.

The NPS sequence is a variant of the Widom 601 sequence. This is now mentioned in the Methods section. We did previously show that *Pf* histone octameres lost their ability to position nucleosomes on genomic DNA fragments (Silberhorn et al. 2016). However, they form rather unstable, but positioned nucleosomes on the NPS containing remodeling templates.

As described above, we now included additional remodeling experiments using AT-rich DNA and *Pf* histone octameres.

5. The authors argue that a major mechanism through which SNF2L activates genes is changes in nucleosome positioning. The nucleosome positions shown in Figure 2D differ between schizont and late ring stage (by the criteria used to show the effect of iKO). Is that due to the expression state of the underlying genes?

The reviewer is correct, the nucleosome positions are changing specifically with the developmental stage of the parasite. This is well documented in a study performed by the Bartfai lab (Kensche et al., NAR 2016), showing a dynamic reorganization of nucleosome positions and ATAC peaks throughout the erythrocytic life cycle. According to our knockout experiments, we can now conclude that *Pf*Snf2L is contributing to these global changes of chromatin structure, thereby affecting gene expression networks, as described in our manuscript.

ED6d suggest that the main positioning clue in the late ring stage is the transition between general AT-rich DNA and GC-rich coding sequences (AUG and atop codon). Interestingly, the positions of the nucleosome at the boundary are hardly affected by KO. The induced knock-out mainly leads to reduced occupancy of nucleosomes outside of reading frames. Does SNF2L affect nucleosome assembly at AT-rich DNA rather than positioning?

Indeed, the stable, high occupancy nucleosomes at TSS, AUG and stop sites remain and are not disrupted. However, it is hard to see in the 2 kbp resolution of the plots, but the nucleosome peak centers are significantly shifted by a few base pairs. Even small movements by about 5bp (half a helical turn of DNA) would fully change the DNA surface of the nucleosomal DNA and potentially change the interaction landscape for DNA binding factors.

The nucleosomes at these boundaries are maintained but their precise positioning is altered. The biggest changes are observed adjacent to the boundary nucleosomes, altering again the broadness of nucleosome-free regions and thereby changing the accessibility of the DNA for potential DNA binding factors.

Whether the changes in the nucleosomal landscape, affecting the “just in time” transcriptional regulation are an effect of direct nucleosome movement or nucleosome displacement/re-assembly cannot be concluded from our experiments. ISWI-type enzymes (and *PfSnf2L*) have been shown to possess all of these activities.

We now show that *PfSnf2L* is able to remodel AT-rich nucleosomal DNA templates (see above), but we have not established competitive remodeler-dependent nucleosome assembly assays to ask whether *PfSnf2L* would preferentially assemble nucleosomes on AT-rich DNA. As *PfSnf2L* preferentially binds to AT-rich DNA, we would speculate that it may do so. However, this issue is not addressed in the manuscript.

6. I did not find the description of the ATPase kinetic experiment (ED 8a) described in the methods. Does NH125 affect *hsSNF2L* in such a purified assay?

The kinetic assay, is now shown in Extended Data Figure 9a and described in the Methods section. NH125 does not affect *HsSnf2L* activity in the biosensor assay and nucleosome remodeling assays (data not shown).

We included a new set of biophysical data showing that NH125 does not interact with human *Snf2H* (Extended Data Figure 10a, b and c). The effects of the inhibitor on *PfSnf2L* and the biophysical experiments are described in the Supplementary Information. Details are also given below.

Minor issues

7. Figure 1c: To which extent are the ATPase activities stimulated by free DNA? A conclusion about ‘nucleosome-stimulated’ ATPase can only be done relative to free DNA. This is important since nucleosome density varies with CG-content in the plasmodium genome and the enzyme requires linker DNA (ED2e) to bind to nucleosomes.

We have performed a detailed analysis of the ATP-dependent activity of *PfSnf2L* compared to *HsSnf2L*. We do not include these datasets here, as there are mechanistic and regulatory differences between the two enzymes that are not in the scope of this manuscript. We used the simplistic term “nucleosome stimulated ATPase”, which is used for ISWI-type enzymes in general. Indeed, the *PfSnf2L* ATPase is stimulated by nucleosomes, like for *HsSnf2L*. However, the *PfSnf2L* ATPase is also activated by DNA alone, in contrast to *HsSnf2L*. But, *PfSnf2L* is an H4-tail-dependent remodeling enzyme like *HsSnf2L*, meaning that the ATPase is already active in the presence of DNA, but requires the H4-tail for active remodeling. The reason for this is the lack of the auto-inhibitory domain of *PfSnf2L* compared to *HsSnf2L*. This is a mechanistic difference that will be addressed in detail in another manuscript.

8. Figure 3D: I do not understand how the NFR width is determined (and could not find the description in the methods).

The NFR width was defined in the method section as follows: “Distance between +1 and -1 nucleosome as detected by the nucleosome dynamics txstart function was used as NFR

width estimation.” We have now moved the description of the method to the Supplementary information.

Request of the Editor – “anything to strengthen the SNF2L inhibitor part would be appreciated”.

In addition to the experiments requested by the reviewers, we included an additional biophysical analysis of NH125 binding to *Pf*Snf2L, *Plasmodium vivax* Snf2L and human Snf2L. The biophysical experiments are described in the Supplementary Information; we included an additional Extended Data Figure 10 and described the results in the manuscript lines 262-271).

In short, we observe the *in vitro* binding of NH125 to recombinant *Pf*Snf2L with about 500 nM affinity, using Microscale Thermophoresis (471nM) and Initial Fluorescence (591 nM) quantification (new Extended Data Figure 10a). This is in good agreement with the even higher affinity or efficacy of NH125 *in vivo* that is observable at 100 to 200nM, depending on the model system used (new Figure 4e and f; new Extended Data Figure 11c, d). We now show that NH125 is highly specific for the *Plasmodium falciparum* enzyme, binding only at concentrations above 20 μ M to the related *Plasmodium vivax* Snf2L protein, and no detectable binding to the human protein, as analyzed by Differential Scanning Fluorometry (nanoDSF) (new Extended Data Figure 10b). Differential Scanning Fluorometry experiments show that NH125 does not bind to the ATP or DNA binding site of the enzyme, still allowing the binding of ATP and DNA. However, binding of NH125 to *Pf*Snf2L results in structural changes and aggregation of *Pf*Snf2L, as documented by Dynamic Light Scattering (DLS) analysis (new Extended Data Figure 10c). NH125 is an allosteric inhibitor of the ATPase domain without affecting substrate binding.

Furthermore, we attempted to generate NH125-resistant parasites by drug exposure similar to (Singh Sidhu et al., JBC 2007, PMID: 17110371) for more than 6 months (data not shown). In any of the used conditions (low/high NH125 for short/long-term exposure), no viable parasites could be detected after the initial toxic effect. The data suggest that compensatory mutations do not occur easily, strengthening the usefulness of *Pf*Snf2L as an efficient drug target.

Together with the gametocytogenesis assays, we now show that our inhibitor has a unique efficacy in that it inhibits at the same time the erythrocyte cycle and gametocytogenesis, without acquiring resistance.

Reply to the reviewer's comments

We prepared a point-by-point rebuttal and prepared a revised manuscript to clarify issues raised by reviewer 1.

Our rebuttal text and the changes in the manuscript are shown in green color. The reviewer's text is given in black color.

R1:

The revisions further improved an already strong manuscript.

Unfortunately, the my main question/concern remains:

Is PFSNF2L, a specific regulator of gene expression that controls or "governs" the activation and silencing of specific *P. falciparum* expression programs in a sequence specific manner OR is it an essential component of the general machinery required for the silencing and activation of most genes?

The former argument is the case. PfSnf2L is a specific regulator required for life cycle progression and differentiation. Our analysis clearly demonstrates that deletion or inhibition of PfSnf2L affects only the regulation of stage-specific genes. The remaining genes are unaffected, suggesting it is not part of a general machinery (see our detailed response below).

The second argument for a general function rather than a specific one, is the large number of differentially expressed genes observed in the iKO.

In the rebuttal, the author state that they observed differential gene expression for 75% of genome's 5600.

This indicates that all or nearly all transcription is affected thereby strongly indicating a general role in transcription, not a specific one, and speaks against a primarily sequence-dependent specific role.

We start by addressing the second argument of the reviewer, as this is the main point. The other points are addressed below.

We realized that the explanation of our data and the plots in the manuscript were not clearly described. We re-wrote this paragraph and included updated Figures.

We made not clear enough that upon PfSnf2L disappearance, not suddenly 75% of the genes are de-regulated. This is only the case if we sum up all de-regulated genes throughout the whole erythrocytic life cycle. Almost all genes are once up- or down-regulated throughout the life cycle. As we postulate that PfSnf2L is a just in time regulator of activation/repression, it would affect this process for all genes at the given stage.

To clarify the selective and stage-specific regulation of PfSnf2L, we display RNA expression in another format, showing pie charts in Figure 3b, including the number of genes being affected at different time points of the erythrocytic life cycle. Furthermore, to clarify the grouping of stage-specific genes and to visualize the loss of "just in time" regulation by PfSnf2L, we included additional supplemental Figures, displaying our data in the more detailed Extended Figures 8b, c and d.

As we describe in the manuscript, only stage-specific genes are the target of PfSnf2L regulation. Only the genes being activated or repressed in a specific phase of the life cycle exhibit misregulation in this phase, while the remaining genes are not affected.

We follow gene expression changes throughout the erythrocytic life cycle after deletion of PfSnf2L, taking samples every 5 hours. Taking together all RNA-seq time points we observe a misregulation of a total of 4200 genes over the whole 48-hour asexual life cycle.

When looking at the RNA-seq time points individually, especially at early time points, the number of misregulated genes is **much lower and specific**. *E.g.* when knocking out PfSnf2L at T0, we see misregulation of only the genes regulated at T45 (217 downregulated genes and 17 upregulated genes, Figure 3b), correlating with the disappearance of PfSnf2L. The affected genes are highly enriched in genes encoding exported proteins (“exportome”). This gene set is known to be activated at this developmental stage and peaks in early ring stages. Importantly, when PfSnf2L is deleted at later time points, we see the specific misregulation of genes known to be regulated at later stages (Extended Data Figure 8), correlating with the disappearance of PfSnf2L. (Figure 3b). The gene set affected at this later time point is enriched in genes encoding factors involved in DNA replication. These genes are normally upregulated during the transition from the ring to the trophozoite stage. In contrast, genes encoding secretory organelle- and mobility-related factors, that should be repressed at that time point, appear upregulated.

In summary, our data demonstrate that PfSnf2L specifically regulates gene sets that are activated or repressed during life cycle transitions. These results are fully validated by specifically inhibiting PfSnf2L with NH125, which results in the misregulation of the same, stage-specific groups of genes (Figure 4d).

The updated text in the manuscript reads now as follows:

“ Time-resolved transcriptomic analysis of KO-parasites reveals a delayed regulation of gene expression, correlating with the disappearance of *PfSnf2L* (Extended Data Fig. 8a, b). Most *Pf* genes show a stage-dependent expression pattern, being activated or repressed at specific time points of the erythrocytic life cycle. These genes were grouped into four stage-specific gene clusters, showing expression peaks in early ring (1417 genes), late ring (1226 genes), trophozoite (1421 genes), or schizont/merozoite (1389 genes) stages (Extended Data Figure 8c, Extended Data Table 3). The loss of *PfSnf2L* leads to delayed activation of genes being activated in this respective stage, and genes being turned off in this stage are similarly delayed in their repression (Extended Data Fig. 8d). These changes in timing fidelity globally slow down the circular trajectory of gene expression in the erythrocytic life cycle of the parasites, with the first effects being visible approximately 40 h post KO induction (Fig. 3a and Extended Data Fig. 8b). Considering all stages throughout the asexual blood cycle, distinct stage-specific sets of genes are differentially expressed (DEG) depending on the early (0h), or late (24h) induction of KO, with about 200 DEGs in the schizont stage, mainly early-ring-specific genes, and roughly 2000 in the ring and trophozoite stages (Fig. 3b). The data shows that *PfSnf2L*-KO does not affect the expression of specific gene categories, but a loss of the remodeling enzyme rather impacts genes undergoing activation or repression at a given developmental stage. This indicates that *PfSnf2L* is required to maintain proper timing of regulation of the entire parasite transcriptome.”

First, the fact malaria parasites only encode a single orthologs from each family of ATPase remodelers (Fig. S1a) and that many components of remodeling complexes are absent (line 50) suggests that only core functions are conserved without the functional diversification of complexes that enables specialized regulation of specific genes sets.

This is a highly speculative comment, especially since the regulation of chromatin remodeling enzymes and their targeting is also unclear for other eukaryotic remodelers and may still be open to many unappreciated functional roles. For example, there are several examples for RNA mediated targeting and regulation of remodelers (*e.g.* NoRC, Swi/Snf and Mi2). Post-translational modifications and the formation of alternative multiprotein complexes are common themes, creating remodeling complex versatility, which can also be envisioned for the singular PfSnf2L. Furthermore, unpublished studies by the Laengst lab show that the interaction of remodeler with specific transcription factors changes the functional activity of these enzymes. Therefore, many mechanisms can be envisioned to differentially regulate gene networks.

Importantly, our pulldown experiments show several uncharacterized Plasmodium-specific proteins, that could potentially interact with PfSnf2L to form *Pf*-specific complexes, which will be an exciting new field of investigation.

Furthermore, the *Pf* genome is 1/150th the size of the human genome, and 1/4 of the number of protein-coding genes are present. Therefore, it can be easily envisioned that 1/3 of the ISWI remodelers compared to human cells are sufficient for gene-specific regulation in *Pf*.

The third argument against a specific role in just in-time transcription, is that the transcriptional program underlying asexual replication is able to advance even in iKO parasite, though somewhat delayed (Fig 3a., Fig.S7c).

While we appreciate this argument, there are several explanations for this observation. First, as can be seen for many so called “essential” factors, a high degree of redundancy can often be observed, leading to a partial complementation upon deletion. As one would expect, not all transcriptional processes will be inhibited upon deletion or inhibition of PfSnf2L.

Therefore, a certain degree of adaptation of the whole gene regulatory network (GRN) can be expected. GRNs have been demonstrated in other eukaryotes to have a high degree of variability.

However, as we clearly demonstrate, there is a delay of the transcriptional program in the absence of PfSnf2L, ultimately resulting in parasite death. Therefore, although the potentially stochastic activation or repression of some “correct” genes within the GRN can be still observed, our data and interpretation fully support the role of PfSnf2L as a transcription regulator that is responsible for the correct (and non-stochastic) just-in-time regulation of genes.

Finally, the rate of sexual commitment is clearly altered in iKO parasites. However, since sexual commitment requires the activation of two otherwise silent regulators (*gdv1* and *ap2-g*) and loss of PfSNFL broadly alters gene activation and repression generally (Line 164).

As described above, PfSnf2L does not broadly alter gene activation. The reviewer seems to have misunderstood our extensive re-analysis of commitment versus conversion. We stated in the manuscript:

“To differentiate whether sexual commitment or conversion is affected during gametocytogenesis, we used a dual reporter line (Extended Data Fig. 12, 13) that enables independent quantification of the two processes in the same experiment. Consistent with the role of PfSnf2L in **regulating “just-in-time”** transcription, sexual commitment is **not affected** at 200 nM NH125. In contrast, the **sexual conversion rate is significantly reduced** at this concentration, indicating impaired regulation of early gametocyte development (Fig. 4f, Extended Data Fig. 11e, f).”

Therefore this is not a valid argument, since commitment is **not** altered. Instead, parasites are unable to differentiate during early gametocyte development, since the activation and repression of stage-specific genes required during this phase of the sexual cycle fails in the absence or upon inhibition of PfSnf2L.

In our revision we will clarify this in more detail to avoid confusion of commitment vs conversion.

Based on these observations, SNF2L appears to be a general regulator of gene expression rather than specifically "governing" key steps.

Every cog in clock may be essential to its function but that doesn't mean that each cog "governs"/regulates what time it shows. Otherwise, being a "regulator" becomes meaningless.

While we agree that a knockout of essential, global factors of transcription would also result in cell death, it is unlikely that the same phenotype would be observed. As summarized above, we show that PfSnf2L is a **stage-specific regulator of gene expression**. In contrast deletion of a general factor such as RNA Polymerase II would result in a global change and likely in total absence of transcription and immediate cell death, independent of gene regulation or stage specificity.

In this manuscript we characterized the molecular function of PfSnf2L in vitro and in vivo, showing the essential and specific regulatory function of a chromatin remodeling enzyme. We have created a lead compound that specifically inhibits PfSnf2L function and matches the transcriptional misregulation of PfSnf2L in vivo, thereby specifically killing Plasmodium falciparum in the erythrocytic life cycle and also targeting gametocytogenesis.

R2: The authors have addressed my comments and I congratulate them on a beautiful study.

R3: The authors extensive revised their manuscript in response to the comments of editors and reviewer. They clarified the confusion revolving around the nomenclature and characterized the biochemical remodeling reactions with Ph histones and on AT-rich DNA. It is fairly obvious, that classical remodeling assays yield only limited information with these atypical nucleosomes. Given the documented expertise of the authors (also highlighted in

previous work), I assume that the results are 'as good as it gets'.

I appreciate the wealth of new data on the developmental effects of KO and inhibitor treatment and on the binding of the inhibitor to the enzyme. While the biochemistry is consistent with the in vivo effects on nucleosome positioning and differential gene expression, it is these in vivo data (including inhibitor effects) that make the manuscript a nice publication that is suitable for 'Nature'.

I have no concerns about publication of the manuscript.